# Carbonate complexation enhances hydrothermal transport of rare earth elements in alkaline fluids

Marion Louvel [1,4 ✉], Barbara Etschmann[2 ✉], Qiushi Guan[2 ✉], Denis Testemale[3 ✉] & Joël Brugger [2 ✉]

Rare earth elements (REE), essential metals for the transition to a zero-emission economy, are mostly extracted from REE-fluorcarbonate minerals in deposits associated with carbonatitic and/or peralkaline magmatism. While the role of high-temperature fluids ($100 < T < 500\,°C$) in the development of economic concentrations of REE is well-established, the mechanisms of element transport, ore precipitation, and light (L)REE/heavy (H) REE fractionation remain a matter of debate. Here, we provide direct evidence from in-situ X-ray Absorption Spectroscopy (XAS) that the formation of hydroxyl-carbonate complexes in alkaline fluids enhances hydrothermal mobilization of LREE at $T \geq 400\,°C$ and HREE at $T \leq 200\,°C$, even in the presence of fluorine. These results not only reveal that the modes of REE transport in alkaline fluids differ fundamentally from those in acidic fluids, but further underline that alkaline fluids may be key to the mineralization of hydrothermal REE-fluorcarbonates by promoting the simultaneous transport of (L)REE, fluoride and carbonate, especially in carbonatitic systems.

[1] Institute for Mineralogy, WWU Münster, Münster, Germany. [2] School of Earth, Atmosphere & Environment, Monash University, Clayton, Australia. [3] University Grenoble Alpes, CNRS, Grenoble INP, Institute Néel, 38000 Grenoble, France. [4] Present address: Institut des Sciences de la Terre d'Orleans CNRS-UMR7327, Orleans, France. ✉email: marion.louvel@cnrs-orleans.fr; barbara.etschmann@monash.edu; qiushi.guan@monash.edu; denis.testemale@neel.cnrs.fr; Joel.Brugger@monash.edu

Meeting the increasing demand for rare earth elements (REE) requires a fundamental understanding of the geological processes that enable REE concentration in the Earth's crust. Carbonatite and peralkaline intrusive complexes (nepheline syenites and granitic suites), as well as their alteration and weathering products (e.g., fenites, placers, and ion adsorption clays), are the primary sources of REE for economic extraction. In general, carbonatite-related systems are enriched in light-REE (LREE: La to Sm), whereas alkaline syenite and peralkaline silicate complexes often display heavy-REE (HREE: Eu to Lu, +Y) enrichment[1–3]. REE primary magmatic enrichment stems from a combination of processes that include liquid–liquid immiscibility and extreme fractional crystallization[1,4–7], and leads to the concentration of REE in minerals such as apatite, pyrochlore, monazite/xenotime, REE-(fluor-)carbonates, or eudialyte, depending on melt composition. Subsequently, late magmatic-hydrothermal fluids can extract REE from magmatic phases and re-concentrate them in newly formed hydrothermal minerals such as monazite/xenotime, fluorapatite, or fluorcarbonates (bastnäsite, parisite, synchysite); this process is often considered a prerequisite to the formation of ores associated with both peralkaline granites and carbonatites[8–11].

A key question however remains concerning the extent and conditions of hydrothermal remobilization and redistribution of REE in different localities, as the spatial distribution of hydrothermal ores, their mineralogy, grades, and LREE to HREE ratios vary widely[1,10]. Hydrothermal mineralization of REE in carbonatite systems generally results in preferential LREE mobilization over the HREE and the precipitation of strongly LREE-enriched phosphates and fluorcarbonates in veins and breccias. However, several carbonatite-related occurrences also present relative enrichments in the valuable HREE in late-hydrothermal xenotime or fluorapatite (e.g. Huanglongpu district, China; Lofdal and Okorusu, Namibia; Songwe Hill, Malawi; Amba Dongar, India)[7,12,13]. These unusual HREE enrichments have been suggested to result from (1) the preferential leaching of LREE out of the carbonatite magmas and/or magmatic ores[7]; (2) sequential precipitation of HREE then LREE-bearing minerals upon cooling and mixing with meteoritic water[13]; or (3) a combination of primary source and magmatic enrichment processes (e.g., HREE concentration in magmatic calcite by fractional crystallization) followed by non-selective secondary hydrothermal remobilization of the REE[12,14]. In peralkaline syenites and granites, hydrothermal processes have also been reported to promote either the dissemination of HREE through the formation of less concentrated pseudomorph assemblages within the magmatic intrusion (e.g., Illimaussaq and Greenland)[15,16] or their enrichment at the boundaries of said pseudomorphs or in hydrothermal veins (e.g., Thor Lake and Strange Lake and Canada)[17–19] while LREE may be further mobilized and deposited in distal fractures.

Based on the common Cl- and SO4-rich nature of fluid inclusions trapped in gangue minerals[14,20] and available experimental studies and thermodynamic models (see Migdisov et al.[21] for review), REE hydrothermal mobilization and LREE/HREE redistribution in new phases are generally interpreted as the result of the selective solubility of REE minerals in evolving hydrothermal systems: REE complexation in acidic fluids promotes REE dissolution, while pH buffering or the addition of fluorine, phosphates, and carbonates from host rocks promotes precipitation[21–27]. Under acidic conditions, the increased stability of LREE-Cl over HREE-Cl aqueous complexes substantiated experimentally at $T > 200\,°C$ is expected to favor the mobilization of LREE and hence promote LREE/HREE fractionation. In contrast, sulfate is deemed an unselective ligand[22] and the formation of REE-SO4 aqueous complexes generally does not result in significant fractionation among the different REE[14]. This "acidic

model" has been successfully applied to account for the hydrothermal redistribution of REE from LREE-enriched bastnäsite and HREE-enriched zirconosilicates (eudialyte or elpidite) within peralkaline granites and pegmatites prospects (e.g., Strange Lake, Canada; Ambohimirahavavy, Madagascar)[18–20,28,29]. At Strange Lake, it has been suggested that the hydrothermal stage involves preferential LREE leaching as chloride complexes in high-temperature acidic fluids, resulting in relative HREE enrichment of residual, leached minerals. Further fractionation of HREE in secondary phases was favored by the enhanced solubility of HREE-rich zircon and other zirconosilicates in low-temperature acidic F-rich fluids[18,19]. The acidic transport of REE as Cl$^-$, SO$_4^{2-}$, and, to a lesser extent, F$^-$ complexes have also been suggested to account for a variety of late-stage REE enrichment in carbonatites, including the formation of calcite + bastnäsite + barite + fluorite + sulfides veins at the Weishan deposit, China[30] or the replacement of magmatic monazite and bastnäsite by HREE-rich secondary phases at the Dashigou deposit, China[12,14].

However, the generalization of the acidic model to all types of REE enrichments, and notably those associated with carbonatite intrusions, is questionable as it does not account for all field observations. For example, the development of fenite aureoles in and around carbonatite intrusions attests to the circulation of high-temperature ($T < 500$–$700\,°C$) alkali- and carbonate-rich fluids in these systems[11,31]. Furthermore, the abundance of calcite veins in these systems suggests that at least part of the hydrothermal activity may have been buffered to near-neutral to alkaline conditions. Together, these observations call for a better evaluation of the role of alkaline fluids in the REE concentration processes in carbonatitic systems. Both thermodynamic models and in-situ spectroscopic measurements demonstrated that under acidic conditions fluorine acts as a precipitating ligand, triggering near-quantitative precipitation of insoluble REE fluorides rather than contributing to their hydrothermal transport[21,26,32]. This leads to a particular conundrum in attributing a hydrothermal origin to the widespread REE-F-carbonate association found in carbonatitic systems via the acidic model since REE and fluorine cannot a priori be transported effectively in the same fluid. A growing number of experimental studies also suggest the efficient mobilization of REE by high-temperature alkali- and carbonate-rich fluids. Tsay et al.[33] originally demonstrated that high amounts of REE (100–1000's ppm) and especially HREE (Gd, Dy, Er, and Yb) could be dissolved in diluted Na$_2$CO$_3$ solutions (0.7 m Na$_2$CO$_3$) under the high P-T conditions of slab dehydration (800 °C, 2.6 GPa). However, experiments by Song et al.[34] showed that REE strongly partitions into carbonatitic melts over aqueous fluids at 700–900 °C and 100–200 MPa, questioning the ability of carbonate-derived fluids to effectively mobilize REE under typical crustal conditions. More recent experiments by Anenburg et al.[35] appear to support Tsay et al.'s observations, with alkali-, carbonate-, and F-rich fluids enabling REE and especially HREE (Dy) redistribution from crystallizing carbonate melts (>55 wt% CaCO$_3$) into newly formed minerals (i.e., fluorapatite, burbankite, and bastnäsite) and quenched solid solutions between 1200 and 200 °C. Finally, a technical study of REE quantification methods in fluids also reported 10's to 100's ppm REE dissolved in Na$_2$CO$_3$/NaHCO$_3$ solutions to 100–200 °C and 4 MPa[36], further confirming the potential of alkali- and carbonate-rich solutions to carry significant amounts of REE, and potentially scavenge and transport REE in and out of carbonatite intrusions. While the relative HREE enrichments reported by Tsay et al.[33], Anenburg et al.[35], and Kokh et al.[36] suggest that the REE complexes formed in alkali- and carbonate-bearing fluids could be HREE-selective, the actual speciation of REE in alkaline fluids remains unconstrained from such experiments. The only available constraints on REE speciation under neutral to alkaline conditions are from the early thermodynamic models of Haas et al.[37] and Wood et al.[38], which generated thermodynamics properties for REE(OH)$^{2+}$,

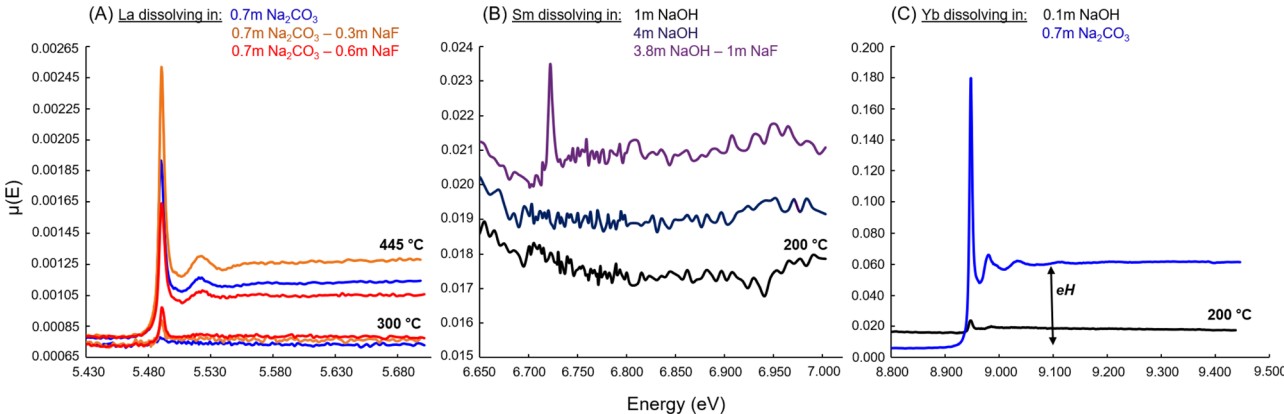

**Fig. 1 Examples of fluorescence spectra used to evaluate La (A), Sm (B), and Yb (C) hydrothermal concentrations.** Enhanced solubility of REE in the hydrothermal fluids is evidenced by an increase of the fluorescence absorption edge jump, which is shown as eH on (**C**). La data were collected at 40 MPa, other REE data were collected at 80 MPa.

REE(OH)$_2$$^+$, REE(OH)$_3$(aq), REE(OH)$_4$$^-$ and REE(CO$_3$)$^+$ to ~1000 °C and 500 MPa via extrapolations from ambient conditions. As demonstrated recently for uranyl carbonate complexing, such semi-empirical extrapolations can be remarkably accurate, but may also fail spectacularly[39]. Thus, current interpretations of hydrothermal mineralization associated with carbonatites remain limited as they can only rely on high-T thermodynamic properties for REE solids (REECl$_3$(s), REEF$_3$(s), REEPO$_4$(s), some fluorcarbonates [2119,24,25]) and aqueous complexes in acidic fluids (REE-Cl, REE-F, and REE-SO$_4$ at pH < 3[21–23,26,27]).

Here, we overcome this limitation by investigating the solubility and speciation of REE in alkaline fluids via in situ X-ray absorption spectroscopy (XAS) measurements to 100–500 °C and 80 MPa. These experiments confirm that carbonate-rich fluids may transport significant amounts of REE under typical hydrothermal conditions and further reveal potential means of transporting REE, F, and other ligands together without the systematic precipitation of insoluble REE fluorides.

## Results

**High-temperature in situ XAS as an experimental window into the hydrothermal behavior of REE.** The study of trace element behavior in hydrothermal fluids is challenging for a number of reasons, including low concentrations and potential back-reactions upon quenching that lead to retrograde precipitation of new mineral phases, possibly obliterating the high-temperature information. Therefore, studies quantifying the solubility of REE minerals, their partitioning between different phases, or the mechanisms that enable their hydrothermal transport have been limited in terms of phases (e.g., the solubility of REECl$_3$(s), REEF$_3$(s), and, more recently, REEPO$_4$(s)) and pH-P–T conditions (pH < 2 and T < 300 °C)[21,24,25,36,40–42].

To overcome quench-related issues, we have taken advantage of a dedicated autoclave equipped with X-ray transparent high-pressure windows that enable the in situ characterization of hydrothermal fluids to 600 °C and 150 MPa via XAS[43]. This approach was used to assess separately the solubility of La$^{3+}$, Sm$^{3+}$, Gd$^{3+}$, Er$^{3+}$, Yb$^{3+}$, and Y$^{3+}$ in alkaline solutions containing varying amounts of OH$^-$, F$^-$, and CO$_3$$^{2-}$ up to 500 °C and 40–80 MPa; and the geometry of Gd- and Yb-carbonate complexes at 200 °C and 80 MPa.

**Concentration trends of REE in high-temperature alkaline fluids.** REE concentrations in high T fluids are a complex function of REE aqueous speciation and the stability of different REE-

bearing phases (REE$_2$O$_3$(s), REE(OH)$_3$(s), REEF$_3$(s), REEPO$_4$(s), or REE-fluorcarbonates) that may form under hydrothermal conditions[21,23–25]. In situ XAS measurements were used to probe the evolution of REE concentrations in the high T fluids as a function of temperature, fluid composition, and pH. To separate the effect of fluid pH from that of different ligands present in solution (OH$^-$, F$^-$, and CO$_3$$^{2-}$), the solubility of La, Nd, Sm, Gd, Yb, and Y oxides (REE$_2$O$_3$) was investigated to 500 °C and 40–80 MPa in (i) single salt solutions (NaF, LiCl) at neutral pH; (ii) NaOH ± NaF ± NaCl solutions at pH > 12; and (iii) Na$_2$CO$_3$ ± NaF solutions at pH ~10 (details in "Methods" section).

Due to a combination of experimental (i.e., low energy of the REE L$_3$-edges) and compositional (i.e., low solubility of REE phases) limitations, concentration values, which are calculated from the amplitude of the absorption edge from transmission XAS spectra, could only be obtained for Yb. For all other REE, we refer to an "apparent" solubility trend. The evolution of REE apparent concentrations with increasing T and different composition is evaluated by the so-called eH value (Figs. 1 and 2), which corresponds to the fluorescence absorption edge jump, i.e., the difference of fluorescence intensity before and after the REE L$_3$-edge that is proportional to the REE concentration in the sample at a given temperature, assuming a fixed photon path from sample to the detector. Additional information about calculated and apparent concentrations can be found in the "Methods" section.

To 500 °C, the apparent concentrations of Sm, Er, Yb, and Y in single salt and NaOH solutions remain at or below the detection limit of our in situ measurements (~10 ppm) (absorption edge is absent or barely visible on Fig. 1). The low concentrations of REE in neutral salt solutions tallies with Pourtier et al.'s[41] study that demonstrated solubility of monazite-(Nd) below ppm levels at 300 °C and 200 MPa. However, the addition of 1 m NaF to a 3.8 m NaOH solution appears to increase Sm solubility slightly, with aqueous concentrations up to an estimated ~30–50 ppm at 200 °C (Fig. 1B). Hence, contrary to acid systems where it acts as precipitating ligand[21,26], fluorine appears to promote REE mobility under basic conditions.

The amounts of dissolved REE (La, Gd, and Yb) further increase in carbonate (±F) solutions (Figs. 1 and 2). However, temperature affects LREE and HREE differently (Fig. 2): Gd and Yb display a retrograde solubility from 100 °C onwards, similar to what we previously reported for Eu, Yb, or Y in acidic Cl- and S-bearing solutions[26,27,44]; on the contrary, La solubility increases suddenly from below detection limit (~10 ppm) at T < 300 °C

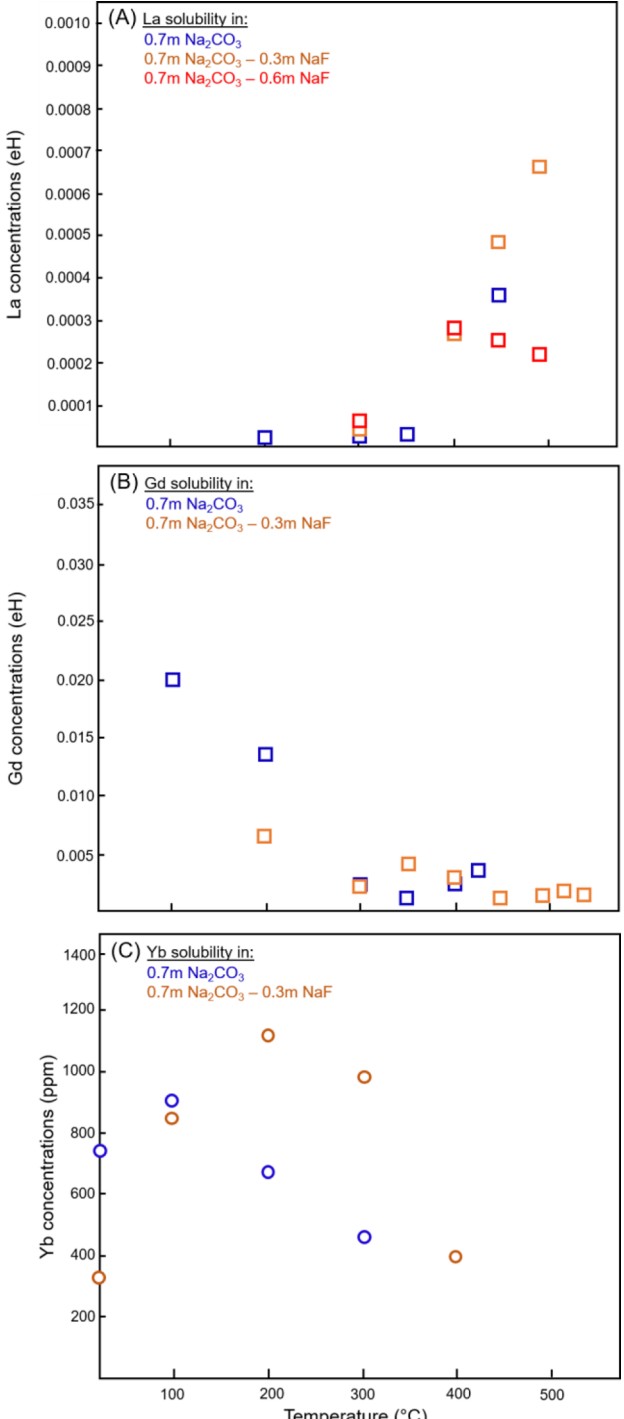

**Fig. 2 Effect of temperature on the concentrations of La(A), Gd (B), and Yb (C) in Na$_2$CO$_3$ ± NaF solutions.** Solubility trends for La and Gd are here represented as the absorption edge height (eH) value from fluorescence spectra. This value is correlated to the concentration of the REE in solution, but may be affected by changes in density and absorption of the different solutions with increasing *P–T* (see "Methods" section). Therefore, it is only an indication of solubility behavior and cannot be used to directly calculate the concentration of La and Gd in the solution. For Yb, absolute concentrations could be calculated from the transmitted spectra, and are reported as ppm Yb in solution.

(e.g., absorption edge is barely visible on Fig. 1A) to more than 10 times the detection limit at 400–500 °C (Figs. 1A and 2A). This is the first time we observe prograde solubility for an REE in our in situ experiments[26,27,44].

In F-free experiments, the maximum apparent concentrations of Gd are estimated to reach 400–600 ppm at 100 °C (Fig. 2B). For Yb, the concentrations calculated from the transmission spectra are close to 900 ppm at 100 °C and decrease toward 400 ppm at 300 °C (Fig. 2C). The concentrations of La at $T > 400$ °C are difficult to assess due to the potential effect of decreasing fluid density on the fluorescence signal (see "Methods" section). Remarkably, the addition of fluorine did not suppress La, Gd, and Yb solubility; on the contrary, the maximum La and Yb concentrations are obtained for a mixed 0.7 m Na$_2$CO$_3$–0.3 m NaF solution. Furthermore, at $T > 400$ °C, the apparent La concentrations become lower in F-rich (0.6 m NaF) carbonate solutions (Fig. 2A), suggesting a "Goldilocks effect" for La hydrothermal remobilization in alkaline fluids, where the optimal conditions encompass moderate amounts of carbonate and fluorine. For Yb, the addition of F is associated with higher concentrations (1100 versus 650 ppm at 200 °C) and delays the solubility drop to 300–350 °C. At 400 °C, 400 ppm Yb are left in the alkaline fluid, about 3–4 times less than the amount of Yb that can dissolve in acidic Cl-rich fluids (0.35 m HCl), but still, a significant amount compared to REE-mineralizing fluids in nature (e.g., up to 7.8 ppm Yb in hypersaline magmatic fluids with up to 80 wt% total salt and homogenization temperatures between 260 and 480 °C[45]). The addition of F does not affect Gd apparent concentrations significantly: it is slightly lower at 200 °C, but eH values close to our detection limits suggest that only tens of ppm of Gd may be retained in both F-bearing and F-free carbonate fluids at T > 300 °C (Fig. 2B).

**Speciation of REE in (CO$_3^{2-}$, F$^-$)-bearing alkaline fluids.** The enhanced solubility of La, Gd, and Yb in (CO$_3^{2-}$ + F$^-$)-bearing fluids compared to pure salt and NaOH solutions suggests that carbonates and/or F$^-$ are more effective ligands than hydroxide for REE transport under alkaline conditions. To test this hypothesis, the structural parameters describing Gd and Yb speciation in the 0.7 m Na$_2$CO$_3$ and 0.7 m Na$_2$CO$_3$ + 0.3 m NaF solutions were extracted from EXAFS spectra collected at 200 °C (Fig. 3 and Supplementary Material S1), where maximum concentrations (500–1500 ppm) were recorded for these elements. The same could not be done for La, due to the decreased signal-to-noise ratio at the lower energy of the La L$_{III}$-edge (5.483 keV, compared to 7.243 keV for Gd L$_{III}$ and 8.944 keV for Yb L$_{III}$).

For Yb, there are obvious differences between EXAFS spectra reported from the CO$_3^{2-}$ ± F$^-$ alkaline solutions and an acidic control solution containing 0.1 m HCl that was measured over similar *P–T* conditions: namely, the first EXAFS oscillation is shifted from ~3.2 to 3.0 Å$^{-1}$ and long-range oscillations (>4 Å$^{-1}$) display a different shape (Fig. 3). More strikingly, these alkaline data are unusual for aqueous fluids in that they exhibit a relatively large peak between $3 \leq R \leq 4$ Å in the phase uncorrected Fourier transform (FT), which underlines the presence of a heavy atom in the second coordination shell. The same is observed for Gd (Supplementary Material S1).

Available spectroscopic studies and theoretical calculations suggest that, at ambient *P–T* conditions, REE form bidentate complexes with the carbonate/bicarbonate ions, with two of the oxygens from the carbonate ion facing the central REE atom, and the third one located at a larger distance in the second shell. The number of carbonate ligands is however disputed depending on

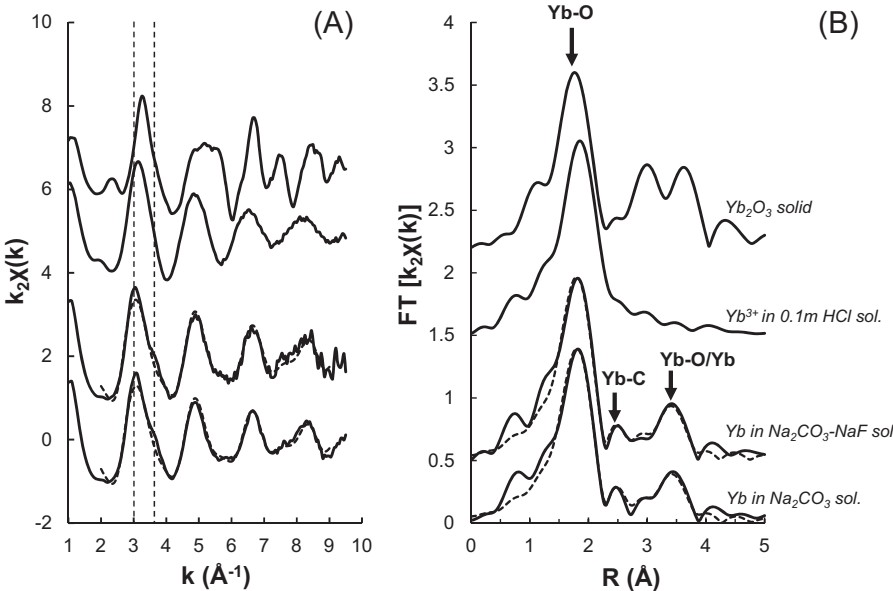

**Fig. 3 EXAFS spectrum (A) and corresponding Fourier transform (B) for Yb in Na₂CO₃ ± NaF solutions at 200 °C and 80 MPa.** The spectrum is compared to that of Yb oxide crystalline compound and Yb dissolved in Cl-rich acidic solution. The fits are reported as a dashed line over the experimental spectra. The two vertical dashed lines on (**A**) underline differences in the shape and position of the EXAFS oscillations between alkaline and acidic solutions. Bold arrows in (**B**) point to the features arising from different scattering paths to oxygen (Yb–O) and carbon (Yb–C) first neighbor atoms or oxygen and ytterbium atoms in the second shell (Yb–O/Yb).

fluid composition and techniques employed: (i) REECO₃⁺ and [REE(CO₃)₂]⁻ have been reported in natural waters based on potentiometric measurements and partitioning experiments[46–48]; (ii) the tetra-carbonate [REE(CO₃)₄]⁵⁻, [REE(CO₃)₄H₂O]⁵⁻, or [REE(OH)(CO₃)₄]⁶⁻ complexes have also been suggested to form in highly concentrated solutions (0.5–5 m (Na,K)₂CO₃), based on UV–vis near-IR absorbance spectroscopy[49–51].

Here, our EXAFS analysis suggests that, in both solutions, the Yb atom is coordinated to eight O atoms located at around 2.29–2.30 Å in the first shell and that only two intermediate C atoms are present at a distance of 2.75 Å (Table 1). The distinctive second shell peak that occurs between $3 \leq R \leq 4$ Å in the FT can be fitted with up to 12 O at an average distance of 4.09 Å (Table Supplementary Material S2). However, the best fits are obtained for a combination of 2–3 Yb and 4–8 O atoms located around 3.85 and 4.1–4.2 Å, respectively (Fig. 3). A similar structure, involving the second shell of 2 Gd and 8 O atoms at 3.87 and 4.28 Å is also fitted for Gd in 0.7 m Na₂CO₃ at 200 °C (Table 1). The formation of polynuclear carbonate species is well-documented for actinide ions[52–54], and polynuclear hydroxide REE complexes have also been described during the hydrolysis of REE with help from organic multidentate ligands such as EDTA[55]. In such contexts, polynuclear clusters are frequently obtained, with two REE sharing two OH⁻ groups. More recently, polynuclear REE-carbonate clusters have also been found to promote the formation of metal–organic framework compounds that are stable to 150 °C, with one CO₃²⁻ bridging up to four REE atoms thanks to its high negative charge and multi-coordination sites[56]. Two examples of such structures were modeled by static quantum mechanical calculations using the Amsterdam Density Functional (ADF) program, based on our EXAFS structural parameters for Gd (Fig. 4). The first one is a hydroxyl-carbonate cluster with the formula [REE₃(CO₃)₂(OH)₄(H₂O)₁₂]⁺, where a central REE carbonate complex is linked to two hydrated REE atoms by four OH groups (Fig. 4a). The second one is a hydrated carbonate polynuclear complex inspired from a uranyl complex[57,58], where each REE atom is surrounded by two REE atoms and three carbonate groups (Fig. 4b); the formula is

[REE₃(CO₃)₃(H₂O)₁₂]³⁺. Taking into account the basic pH of our high P–T fluids and the fact that the EXAFS best fits only account for two carbonate groups, our preferred structure is that of the [REE₃(CO₃)₂(OH)₄]⁺ hydroxyl-carbonate polynuclear cluster (Fig. 4a). The potential precipitation of lanthanite-([REE₂(CO₃)₃·8H₂O](s)) or tengerite- ([REE₂(CO₃)₃·2–3H₂O](s)) like carbonate hydrates in the beam-path can be discarded based on (1) the absence of absorption contrast in transverse transmission scans of the cell or glitches in the XAS spectra that are generally observed upon precipitation under the beam and (2) the presence of only two carbonate groups in the best fits, while lanthanite and tengerite-like structure involve coordination to 3–4 carbonate groups[59]. While the lower REE concentrations found in the presence of F (Gd) or at T > 200 °C (Gd and Yb) hindered EXAFS analysis for these conditions, similarities in the XANES spectra suggest that similar REE hydroxyl-carbonate complexes may be stable to at least 300 °C in both F-free and F-bearing alkaline fluids (Supplementary Material S1).

## Discussion

Experimental studies on the solubility of REE solids in high T (>250 °C) alkaline fluids have been extremely scarce, mostly limited to the fate of the phosphate mineral monazite–(Nd) in hydroxide solutions[41]. While the authors suggested that using the hydrolysis constant from Haas et al.[37] and Wood et al.[38] enables reproducing their observed solubility at 300 °C and 200 MPa, a set of recent experiments by Gisy et al.[25] instead points to a need to revise the thermodynamic properties of REE hydroxyl species for future solubility calculations. Our combined solubility and speciation work now suggest that carbonate may act as a more potent ligand than hydroxide alone and that hydroxyl-carbonate complexes may promote REE transport in alkaline fluids.

The retrograde solubilities of Gd and Yb suggest that HREE transport in alkaline fluids may be limited above 300 °C. As we could not recover any precipitates for analysis (only a fine powder coated the lower piston), it remains difficult to assess whether it is carbonates, fluorcarbonates, or hydroxides phases that control

**Table 1 Structural parameters derived from the EXAFS analysis of Gd- and Yb-bearing alkaline solutions at 200 °C and 80 MPa.**

| Solutions | O1-shell | | | C1-shell | | | REE2-shell | | | O2-shell | | | ΔE0 (eV) | Rfactor | χred |
|---|---|---|---|---|---|---|---|---|---|---|---|---|---|---|---|
| | N | R | σ² | N | R | σ² | N | R | σ² | N | R | σ² | | | |
| *Gd* | | | | | | | | | | | | | | | |
| 0.7 m Na₂CO₃ | 7.9 (7) | 2.39 (1) | 0.009 (2) | 2.1 (6) | 2.85 (3) | 0.003ᵃ | 2.2 (12) | 3.87 (4) | 0.006ᵃ | 7.9 (22) | 4.28 (3) | 0.002ᵃ | 1.02 (50) | 0.025 | 45 |
| *Yb* | | | | | | | | | | | | | | | |
| 0.7 m Na₂CO₃ | 8.1 (3) | 2.31 (1) | 0.012 (1) | 2.1 (9) | 2.75 (3) | 0.006 (5) | 2.4 (7) | 3.85 (2) | 0.001 (3) | 5.5 (23) | 4.14 (2) | 0.003 (3) | 4.1 (3) | 0.014 | 316 |
| 0.7 m Na₂CO₃–0.3 m NaF | 8.3 (3) | 2.30 (1) | 0.011 (1) | 2.3 (4) | 2.75 (3) | 0.008 (5) | 2.4 (8) | 3.86 (3) | 0.0015ᵃ | 7.6 (17) | 4.12 (3) | 0.007 (5) | 3.4 (3) | 0.0184 | 55 |

*N* coordination number, *R* average distance between central atom and neighbors, $\sigma^2$ Debye–Waller factor, $\Delta E_0$ energy shift; *Rfactor* and $\chi_{red}$ goodness of fit.
ᵃThe parameter was fixed for the fit.

HREE solubility under these conditions. In contrast, LREE transport in alkaline fluids may be favored under high *T* conditions, as suggested by the prograde solubility of La from 300 to 500 °C in carbonate-bearing fluids (Fig. 2). While the low energy of La L₃-edge measurements precluded obtaining speciation information through EXAFS analysis, we suggest that hydroxyl-carbonate complexes similar to those reported for the HREE are behind this prograde solubility of La.

Contrary to what is observed under acidic conditions[21,26], fluoride ions do not act as a precipitating ligand in these fluids, at least to moderate concentrations (0.3 m), since F addition is found to increase both La and Yb aqueous concentrations (Fig. 2). Whether this solubility increase is favored by combined fluoride (as $REEF^{2+}$, $REEF_2^+$, or hydroxyl-fluorides $REEF_x(OH)_y$) and hydroxyl-carbonate complexation or the substitution of $F^-$ for $OH^-$ in the $[REE_3(CO_3)_2(OH)_x(H_2O)_y]^{3-x}$ polynuclear complexes, however, remains to be constrained. Unfortunately, similarities between O and F ionic radii and interatomic distances render discrimination between hydroxyl and fluorine neighbors extremely difficult with XAS[60].

The lower La concentration reported in the 0.7 m Na₂CO₃–0.6 m NaF solution above 400 °C (Fig. 2) nevertheless suggests a "Goldilocks effect", where intermediate fluorine contents (0.3 m NaF) promote LREE hydrothermal transport under high *T* conditions whereas higher concentrations (0.6 m) lead to precipitation of secondary REE phases. Though the high *T* precipitates were too few to be recovered and analyzed after the quench, it is likely they were fluorcarbonates that formed through gradual substitution of $F^-$ for $OH^-$ in fluids containing more than 0.3 m NaF. The $[REE_3(CO_3)_2(OH)_x(H_2O)_y]^{3-x}$ aqueous complex we propose from the results of the EXAFS analysis actually shares similarities with the structure of REE fluorcarbonates such as $Na_3La_2(CO_3)_4F$ or horvathite-(Y), $NaY(CO_3)F_2$, where $(REE + Y)O_xF_y$ polyhedra are linked by carbonate groups[61,62]; hence, aqueous polynuclear carbonate complexes could act as a precursor to mineral formation. More common hydrothermal fluorcarbonates such as bastnäsite ($REECO_3F$), synchysite ($REECa(CO_3)_2F$), or parisite ($REE_2Ca(CO_3)_3F_2$) could also directly precipitate following the same mechanism, where F replaces $OH^-$ groups in hydroxyl-carbonate aqueous compounds.

Many field studies have invoked the high amounts of chloride and sulfate daughter minerals recorded in fluid inclusions in carbonatite-related REE deposits to underline the potential role of REE-Cl and REE-SO₄ complexation in the hydrothermal transport and concentration of the REE[14,63,64]. However, the thermodynamic model of Migdisov et al.[21] stresses that such species may only be stable under acidic to near-neutral conditions in F-bearing systems (pH < 3–4 for $REECl^{2+}$ and $3 < pH < 7$ for $REE(SO_4)_2^-$ at 400 °C, 100 MPa). Our results further call for a reevaluation of REE transport mechanisms in carbonatitic systems, as we observed high HREE (Gd, Yb) concentrations in low-temperature (F,$CO_3^{2-}$)-rich alkaline fluids (<300 °C); the apparent prograde solubility of LREE (La) to 500 °C in the same fluids; and characterized the structure of to date unreported hydroxyl-carbonate polynuclear REE complexes.

The nature of fluids circulating in and around carbonatite intrusions is complex, and both orthomagmatic and mixed hydrothermal/meteoritic fluids have been invoked to account for REE mineralization at different sites[7,12,13,63–67]. The magmatic-hydrothermal transition in carbonatites is characterized by alkali-, chloride-, sulfate-, and carbonate-rich fluids being expelled from the crystallizing carbonatite melts. This process ultimately results in pervasive or focused (veins) alkaline alteration of surrounding rocks (fenitization) to different scales (see Elliott et al.[11] for a review). Whether those fluids form gradually as the end products of

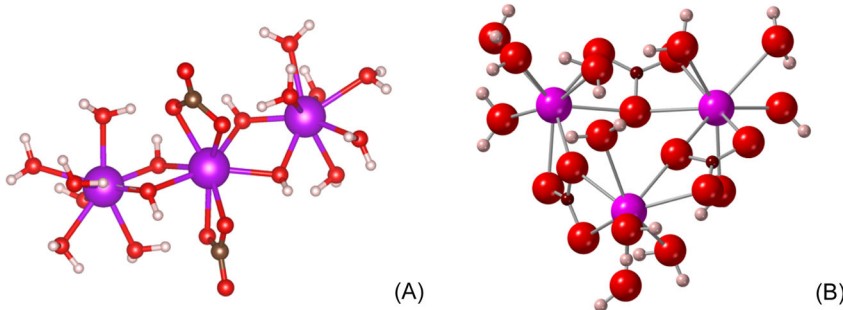

**Fig. 4 Proposed structures for the Gd (hydroxyl-)carbonate complexes.** Structures were obtained by static quantum mechanical calculations using the Amsterdam Density Functional (ADF) program: **A** $[REE_3(CO_3)_2(OH)_4(H_2O)_{12}]^+$, **B** $[REE_3(CO_3)_3(H_2O)_{12}]^{3+}$. The atoms are gadolinium (purple), oxygen (red), carbon (brown), and hydrogen (white).

fractional crystallization of the carbonatite melt; are released as multiple pulses through immiscibility with the melt; lose their C through immiscibility between $H_2O$–$CO_2$ and Cl-rich fluids; or evolve differently as a function of their distance from the source and previous alteration has been a long-standing matter of debate[11,31,68,69]. Experimental constraints on the conditions of fluid production from carbonatites are scarce, but homogenization temperature for fluid inclusions[68–70] suggest that fenitization could occur from 700 to 800 °C down to the sub-solidus of (natro-)carbonatite, i.e., 500–600 °C[71]. While aqueous, high salinity, multi-component fluid inclusions containing halite, nahcolite ($NaHCO_3$), calcite, strontianite ($SrCO_3$), fluorite, and sulfates have been reported at many carbonatite complexes (Kaiserstuhl, Palabora, Fen, Oka, Kalkfeld, Amba Donga, Jacupiranga, and Songwe Hill), their complex mineralogy makes it difficult to assess formation temperatures and original compositions[31,70]. Recently, Walter et al.[70] paved the way for a better evaluation of the high $P$–$T$ compositions of these fluids by using a Margules-type solution model to recalculate the (Na,K)Cl, (Na,K)$CO_3$, and (Na,K)$SO_4$ bulk concentrations of multicomponent fluid inclusions from the Kaiserstuhl carbonatite (SW Germany). They suggest that the fluids derived from the carbonatites at the magmatic-hydrothermal transition contain up to 80 wt% $[(Na,K)Cl + (Na,K)_2CO_3 + (Na,K)_2SO_4]$, and hence constitute brine-melts whose properties are intermediate between those of carbonatite melts and dilute hydrothermal fluids such as those studied here. Raman measurements confirm that $CO_3^{2-}$ and $HCO_3^-$ are the dominant carbon species in the more evolved fluids, demonstrating neutral to alkaline conditions similar to our experiments. Although the fluorine and REE contents of these alkaline brine-melts remain to be quantified, REE-bearing phases such as burbankite, bastnäsite or apatite have been reported as daughter minerals in such inclusions[9,65,72]. Anenburg et al.[35] further suggested that cooling of such fluids could both promote LREE concentration and preferentially remove HREE from the carbonatite body thanks to the formation of burbankite ((Sr,Ba,REE)$_3$(Na,K,Ca)$_3$(CO$_3$)$_5$) and carbocernaite ((Sr,REE,Ba)(Ca,Na,K)(CO$_3$)$_2$), which concentrate the LREE but cannot incorporate large amounts of HREE. Such process could act as a precursor to the typical LREE-enriched carbonatite mineralization as monazite and fluorcarbonate pseudomorphs in carbonatite complexes. While our in situ experiments do not enable us to draw direct conclusions about the solubility of REE in such solute-rich fluids, studies of transition metals indicate little changes in coordination geometry of metals such as Au(I) and Cu(I/II) in the transition from brines to hydrated salt melts (brine-melts)[73]. Hence, we infer that alkali-REE-carbonate clusters could as well promote high REE concentrations in brine melts and the precipitation of burbankite or carbocernaite, in a comparable manner as reported for Zr–O–Si/Na alkali-zirconosilicate clusters and vlasovite in Si and Na-rich fluids[60].

Our in situ constraints on REE in 0.7 m $Na_2CO_3 \pm NaF$ fluids apply directly to purely hydrothermal stages, where early "melt-brines" have been diluted to lower solute contents (<20 wt%) through cooling, fluid-rock interactions and/or mixing with meteoritic fluids. Such fluids may lead to the formation of veins and stockwork zones intruded in the alkaline-carbonatite sequences or host rocks (e.g., deposits of the Mianning Dechang belt, China)[64,74,75]. Late hydrothermal reworking of magmatic assemblages to hydrothermal fluorphosphates and fluorcarbonates such as reported at the Okorusu (Namibia), Amba Dongar (India), or Kangankunde (Malawi) complexes[63,66,67] is also more likely to involve fluids that will not completely leach the carbonate component away, i.e., near-neutral or alkaline fluids where REE may be mobilized as the newly characterized hydroxyl-carbonate complexes or as $REE(SO_4)_2^{-}$[21,63,66]. Our study suggests that depending on temperature conditions, hydroxyl-carbonate complexation should lead to preferential mobilization of the LREE ($T > 300$ °C) or of the HREE ($T < 300$ °C). In the Maoniuping deposit (China), REE are mostly concentrated in late-stage bastnäsite in veinlets that crosscut previous calcite-fluorite-baryte precipitates. According to fluid inclusion studies, these bastnäsites formed at $T \geq 160$–240 °C from sulfate and $CO_2/CO_3^{2-}$ rich fluids[64]. Based on our experiments, LREE-hydroxyl-carbonate complexation under "moderate" temperatures (~300–450 °C) and solubility drop at $T < 300$ °C (Fig. 2) could very well account for the formation of such mineralization. Our results also show that fluorine can be co-mobilized with REE in such fluids. The enhanced LREE mobility as hydroxyl-carbonate complexes at $T > 300$ °C may also explain the residual HREE enrichment of apatite at the Tundulu and Kangankunde carbonatites (Malawi)[76]. On the contrary, REE-$SO_4$ complexation will scavenge REE without significant fractionation among LREE and HREE, and thus lead to the precipitation of secondary phases with LREE/HREE ratios similar to their magmatic source, as recently described by Cangelosi et al.[63] for the Okorusu carbonatite.

Overall, our in situ study highlights new mechanisms that may contribute to the hydrothermal concentration of the rare earth elements and LREE/HREE fractionation in carbonatitic systems. They also reveal that co-transport of all the components present in REE-fluorcarbonate ores is possible under alkaline conditions, whereas fluorine acts as an efficient precipitating ligand under acidic conditions. The circulation of alkaline fluids is however not limited to carbonatite intrusions but its bearing on REE mineralization is not as obvious for nepheline syenite and peralkaline granites. At the Strange Lake deposit (Canada), recent fluid inclusion studies[20,28] highlighted that the earliest hydrothermal stage involves high-temperature (400–500 °C) fluids with pH > 9. These early fluids contain C (trapped as $CH_4$ bubbles in the fluid inclusions), Cl (23 wt% $NaCl_{eq}$), and F (0.2–0.6 wt%), and are enriched in LREE, with La/Yb ratios

~2.3. Based on the absence of carbonate minerals in the early fluid inclusions, Vasyukova et al.[20,28] speculate that early LREE mobilization in the high $T$ Cl- and F-rich fluids are probably due to hydroxy-fluoride complexation similar to that suggested for Zr and Nb[77,78] rather than carbonate complexation such as described here. Alkaline alteration at the Illimaussaq nepheline syenite Complex in Greenland instead leads to preferential loss of the HREE and HFSE from their primary source, as evidenced by their strong depletion in eudialyte pseudomorphs[15,16]. The new hydroxyl-carbonate species identified here may be the first of many unsuspected complexes and advocate for the reevaluation of the role of alkaline fluids in the genesis of REE deposits, via a more systematic characterization of fluid inclusions volatile and trace elements composition, as well as new high $P$–$T$ solubility studies. Such a step may be critical for defining the key chemical and geochemical factors controlling enrichment in the most valuable rare earth (Nd, Pr, and Dy)[2] and developing more effective exploration strategies to sustain REE production into the future.

## Methods

**Experimental set-up**. All experiments were conducted in a dedicated autoclave that enables in situ X-ray absorption (XAS) on high-temperature, high-pressure fluids at the BM-30B beamline at the European Synchrotron Radiation Facility (ESRF, Grenoble, France)[43]. The aqueous samples are contained in a vitreous carbon tube that is placed within a small furnace enclosed in a high-pressure vessel. The vessel is equipped with three 0.8 mm Be windows that enable data collection in both transmission and fluorescence (90°) modes. The aqueous samples are enclosed in the internal vitreous C tube by two vitreous C pistons that can move freely along the tube when pressurizing gas (He) is flushed in the high-pressure vessel or alternatively to accommodate volume expansion upon heating. In the current study, the samples were pressurized to 80 MPa (except for measurements at La $L_{III}$-edge, for which pressure was 40 MPa) and heated up to a maximum of 500–550 °C.

The ESRF is a 6.03 GeV ring and was operated in 7/8 multi-bunch mode, with a maximum current of 180 mA. The FAME beamline is a bending magnet beamline that has been described elsewhere[79]. Energy selection at the $L_{III}$-edges of La, Sm, Gd, and Yb (5.483, 6.716, 7.2432, and 8.944 keV, respectively) was ensured by Si(220) monochromators, with an energy resolution between 0.3 (La) and 0.45 eV (Yb) FWHM. Experiments involving Y were conducted at the Y K-edge (17.038 keV). The monochromatic beam was focused down to $300 \times 100$ µm$^2$ ($H \times V$ FWHM). Incident and transmitted beam intensities $I_0$ and $I_1$ were measured with Si diodes, while the fluorescence radiation was collected with a Canberra 30 element solid-state detector set at 90° from the incoming beam.

X-ray absorption near-edge structure (XANES) and extended X-ray absorption fluorescence spectroscopy (EXAFS) spectra were collected at room temperature, and then in steps of 50–100 °C at high pressure. The aqueous solutions were left to equilibrate for a few minutes after each heating step and spectra were then collected for 1–3 h at each temperature depending on the aim of the measurements (solubility or speciation experiments – see below). The reader is referred to previous publications on Cu, Eu, Pb, Pd, Y, or Yb hydrothermal speciation for further details on the autoclave and beamline capacities at extreme $P$–$T$ conditions[26,27,44,80–82].

**Estimation of REE aqueous concentrations from XAS spectra**. The solubility of rare earth elements in neutral to alkaline solutions was investigated by loading a sintered piece of REE$_2$O$_3$ (Sigma Aldrich©) together with salt, F, hydroxide, and/or carbonate-bearing solutions. Carbonate solutions were prepared by dissolving requisite amounts of Na$_2$CO$_3$ in deionized water. Hydroxide solutions were prepared by diluting requisite amounts of NaOH, 50–52% Sigma-Alrich) in deionized water. When present, F and Cl were added as NaF and NaCl/LiCl (Sigma Aldrich/ACS, ≥ 99.0%). All investigated compositions are summarized in Table S1. The solubility of Yb was determined from the amplitude of the absorption edge Eh at each temperature condition. As our experimental set-up involves a fixed X-ray path, the amplitude of the absorption edge as measured in transmission spectra is directly proportional to the concentration of the excited atoms ($C_i$ in mmol) according to Beer–Lambert's law

$$Ci = \frac{eH}{\Delta\sigma_i . M_i . x . \rho_{sol.}} \quad (1)$$

where $\Delta\sigma_i$ is the change of the total absorption cross-section of the element $i$ over its absorption edge (cm$^2$ g$^{-1}$), $M$ the atomic weight (g mol$^{-1}$), $x$ the optical path length through the sample (cm) and $\rho_{sol.}$ the density of the aqueous fluid at $P$ and $T$ (g cm$^{-3}$).

For La, Sm, and Gd, the strong absorption of the signal by the experimental set-up (air between detectors and autoclave + Be windows) and sample (4 mm of aqueous solution) at the low $L_{III}$ edge energies (5.483–7.243 keV) however prevented collecting XAS spectra in transmission necessary to calculate their aqueous concentrations based on Beer–Lambert's law. Instead, we propose to use the absorption edge jump Eh from fluorescence spectra (calculated with Athena program[83]) as an indicator of "apparent" solubility trends. For a fixed photon path

(i.e., autoclave and fluorescence detectors are kept at the same position throughout the experiments), the fluorescence signal is correlated to the concentration of the excited atoms, but also the density and the absorption of the different solutions at different $P$–$T$ conditions.

To evaluate the effect of density and absorption on the amplitude of the fluorescence absorption edge jump, we reported in Figure S2 the eH values for reference solutions of 1 wt% La in 5 m Cl$_{tot}$, 200 ppm Gd + 200 ppm Yb in 0.1 m HCl and 8500 ppm Yb in 0.1 m HCl that were collected from 25 to 450 °C and pressure between 40 MPa (La) and 80 MPa (Gd, Yb) during the same beamtime as our experimental data. For all compositions, eH values remain stable within error while fluid densities decrease from 1.01–1.2 to ~0.7 g cm$^{-3}$ with increasing temperature to 400–450 °C. Thus, the eH value can be considered to be directly proportional to REE concentration for such conditions and the average eH value of ~0.0081 for the 200 ppm Gd + Yb solution was used to evaluate Gd concentration in the carbonate-bearing solutions at 100–200 °C (400–600 ppm). The eH value for the La solution at $T$ > 450 °C could not be estimated due to the onset of brine-vapor separation. In the case of the Gd and Yb solutions, eH values drop significantly as the density decreases toward 0.5 g cm$^{-3}$ (supercritical conditions). The decrease of eH is however time-dependent at both 400 and 450 °C, suggesting a decrease of Gd and Yb solubility associated with precipitation of Gd and Yb solids, rather than a direct effect of density on the fluorescence signal. While an increase of fluorescence signal under supercritical conditions such as those we report for La in carbonate-bearing fluids is most likely related to an increase in their concentration, it remains difficult to assess the effect of decreasing density below 0.6 g cm$^{-3}$ on the fluorescence signal. Thus, we do not provide an estimation for La concentration at $T \geq 400$ °C.

**Speciation analyses**. The stability of Gd and Yb hydroxyl-carbonate complexes in 0.7 m NaCO$_3$ and 0.7 m NaCO$_3$–0.35 m NaF solutions was investigated simultaneously to solubility.

The XAS spectra were collected from −200 to +800–1000 eV across the corresponding REE $L_{III}$-edges. (6.716 and 8.944 keV). XANES and EXAFS data were analyzed with the HORAE package[83], using FEFF version 9. After normalization and background removal, the EXAFS oscillations were fitted, using a starting model of Gd and Yb(CO$_3$)(OH)[84]. The $k^n$-weighted data (n = 1,2,3) used in the fit ranged from 2 to 9 Å$^{-1}$. The fitting was done in R-space over the range 1.0 to 4 Å. EXAFS transmission signals were noisy and thus fluorescence data were used for the analyses. Aqueous references of Gd$^{3+}$ and Yb$^{3+}$ in 0.1 m HCl were used to determine the amplitude reduction factor S0$_2$ (1 for both elements). Fitted structural parameters include the coordination number $N_i$, interatomic distances $R_i$, the mean-square relative displacement Debye-Waller factor $\sigma^2_i$, the energy shift $\Delta E_0$. The fits were performed simultaneously with $k$-weighting of 1–3 so as to diminish correlations between N and $\sigma^2$.

**ADF calculations**. The geometries of the Gd cluster proposed in Fig. 4 were optimized by static quantum mechanical calculations using the ADF program[85]. The ADF program implements density functional theory. ß Atoms were described with a triple-zeta basis set with polarization functions[86]. Electrons were performed with zero-order regular approximation and BP86 functional[87,88] under generalized gradient approximation. The conductor-like screening model[89] was employed to describe the long-range solvation field of the aqueous solution. The dielectric constant was set as 38.32 to represents the aqueous environment of 200 °C[90]. The solvent radius of 1.93 Å was chosen. The geometry was optimized without symmetry constraints. The optimized coordinates of the Gd-CO$_3$$^{2-}$ cluster are summarized in Table S3. The optimized Gd–O, Gd–C, and Gd–Gd distances are 2.45, 2.90–2.92, and 3.81–3.64 Å, respectively, close to the fitted EXAFS values (Table 1).

## Data availability

The raw XAS spectra for Yb and Gd in 0.7 m Na$_2$CO$_3$ solutions at 200 °C and 80 MPa are available in the H2020 European SShade Solid Spectroscopy database (https://www.sshade.eu).

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

## Acknowledgements
This contribution was made possible by regular access to the CRG beamline BM30-B granted by the ESRF and SOLEIL synchrotrons. M. Louvel acknowledges funding under the EU H2020 Marie Curie-Skłodowska Action (Individual Fellowship 797145 "REESources" to M.L.). The authors also want to thank A.E. Williams-Jones and A. Migdisov for useful discussions. The DFT calculations in this work were supported by resources provided by the National Computational Infrastructure (NCI) supported by The Australian Government (DP190100216 to J.B.).

## Author contributions
The experiments were designed and conducted by M.L., J.B., B.E. Data analysis was conducted by M.L. and B.E., and A.D.F. calculations by Q.G.D. Testemale prepared the experimental and analytical setup and contributed to data collection. M.L., J.B., B.E., Q.G., and D.T. all contributed to the paper.

## Funding

## Competing interests
The authors declare no competing interests.
