## [Peer Review File · Nature Communications]

REVIEWER COMMENTS

Reviewer #1 (Remarks to the Author):

This study presents new a series of hydrothermal experiments and in situ X-ray absorption spectroscopy for the solubility of REE in alkaline carbonate-fluoride-bearing aqueous fluids. The authors aim at relating their experimental results to the transport and mineralization process of REE in mineral deposits in natural systems. The manuscripts compares previous experimental work done mainly at low pH and < 300 °C to their new results conducted at elevated temperature up to 500 °C in alkaline fluids and try to draw conclusions for carbonatite and alkaline hydrothermal-magmatic systems.

Overall, I am excited that the solubility of REE was measured at elevated pressure and temperature, and that we are finally making advances on the speciation, not only in acidic environment, but in alkaline aqueous fluids, which to my opinion certainly play a key in natural REE mineral deposit ore formation. The authors seems to have identified the importance of carbonate complexes in alkaline fluids using spectroscopic analyses. This is a potentially important finding.

On the other hand the manuscript has some problems concerning some of the experimental data interpretations and their presentations in the text, and I encourage the authors to address these to target a broader audience and make it easier to follow. I added detailed comments in the annotated pdf. Here is a summary of the main points:

1) The introduction is somewhat difficult to follow and needs reorganization to better link the problem between our knowledge (or lack thereof) of thermodynamic properties of REE speciation and laboratory experiments vs. observations in natural REE deposits. There are some affirmative sentences without references to back up the ideas presented. Also a few of the references are not adequately cited including in the discussion and there are many typographical errors in the manuscript. I found the same problem in the discussion section where the authors discuss natural systems starting lin 296.

2) In the results section, I suggest to be more explicit in the type of experiments that were used, perhaps even have a short paragraph that explains briefly the advantages of using in situ XAS in geosciences. There are also several places where temperature/pressure or condition of the experiments are not listed. I understand that the details go into the methods section, but it is difficult to separate some of the conditions. I also added some comments about this in the methods sections as the authors mention solubility vs. speciation experiments there but not in the text, and need to clarify the difference (analytical, experimental, or interpretation of spectra).

3) The presentation of experimental results needs to be improved. I have not found any mention in the results explaining the rationale behind the initial setup of the fluid compositions used in the experiments, i.e. lines 116-119. Why did the authors exactly choose these compositions, did they try to isolate the control of REE complexation on REE solubility? The result section also clearly needs to state that the experiments measured the solubility of REE oxides, and explain possible secondary minerals that can form as a result of the presence of fluoride and carbonate in the fluid.

To me the solubility variations seen in Figure 2 can stem from several competing reactions including: i) REE complexation, ii) solubility of the REE solids that were used in the experiments, iii) the stability of secondary REE minerals. All 3 competing reactions depend on pH and temperature and ligand activity. The results only discuss total solubility of REE and the identification of possible REE carbonate aqueous complex in a few of the experiments. It would be beneficial for the reader to clarify that these processes are investigated and which experiment is controlled by which process.

4) Any possible carbonate or fluoride phases that could have formed in these experiments? How would that affect the measured spectroscopic signals? Why were these solids not measured after quenching the experimental fluids? Couldn't they yield valuable information on the experiments?

5) I was confused in the discussion about which REE aqueous species was identified based on which evidence. Figure 4 and text in the results section indicates that a Gd hydroxyl-carbonate species was identified using ADF calculations. The experiments indicate also a possible role of F, although as said above, some experiments show the precipitation of fluoride/fluorocarbonates, which have not been identified. However, the discussion then mentioned (hydroxyl-)carbonate complexes (line 273), perhaps Sm hydroxy-fluoride complexes (lines 321-322), and finally a new (hydroxyl-, fluoride-)-carbonate species (line 388). This needs to be better explained or clarified.

5) A follow up on point 4), the results are not always conclusive, i.e. many of the experiments were below the limit of detection. A big question mark to the reader a bit less familiar with the spectroscopic methods could be: why are all the REE-OH experiments below the limit of detection, or are they not detectable at all at the measured conditions? Can the authors perhaps state whether any of their setup could be changed to better detect these species using the synchrotron, or is it possible to perhaps use higher REE concentrations in the starting fluids for these experiments?

Overall, I see the effort the authors have undertaken to measure the solubility of REE at difficult conditions. While these are important new results, the points above need to be clarified to enable future research to build on these findings. I hope my comments will help improving the manuscript,

Alexander Gysi

Reviewer #2 (Remarks to the Author):

I love the paper by Louvel et al because it demonstrates what we inferred indirectly from our own experiments (Anenburg et al 2020): that alkalis in alkaline systems make REE (and especially HREE!) very soluble. I also love the paper because it shows rather convincingly that we we wrong about the mechanism. It is not directly complexing by alkalis, but rather (as I understand) that alkalis make carbonate available for the REE. It's always interesting when people find you are wrong..!

In terms of novelty and significance, I would definitely recommend this paper be published in NComms. The more nails we can put in the coffin of "acidic mobility of REE", the better.

That said, I still have some issues with the paper that I believe need to be addressed before acceptance.

1. The paper attempts to enhance the significance of their finding that alkaline fluids are responsible by emphasising the importance that others are giving to the acidic model. The issue is that no one actually found evidence for acidic fluids in their carbonatites, they just assume that this is the case, based on the (correct but misinterpreted) works of Migdisov and Willy-Jones. This current paper is already important without repeating those misinterpretations and fallacies made by others. As someone who is deeply involved in the carbonatite community, I would say that people don't support the acidic model because they actually see evidence for acidic fluids in their rocks, they cite it because they see hydrothermal mineralisation in their rocks and they need to cite something that justifies it (often without actually reading or understanding the papers they are citing!).

2. The magma-hydrothermalfluid dichotomy is inherited from the science of silica magma systems. Things aren't so clearly separated in carbonatite systems, where very often there is a continuous transition (without exsolution or phase separation) from carbonatite magma to saline brines, both of which can carry REE. You show retrograde solubility, that the REE contents drop at high temperature. But this is probably not too relevant to natural systems, where at those high temperatures the carbonatite magma still exists, and acts as the REE carrier.

3. Following on the above, there needs to be a clear separation in the text between silicate systems and carbonatite systems. Those are two completely different chemical environments.

4. In carbonatites, the REE mineralisation is often already spatially locked in place in the primary mineralisation from the carbonatite itself. However, those are initially assemblages of alkali minerals like burbankite and carbocearnite, which are very soluble once the Na is removed from the system in later fluids. So the minerals recrystallise insitu to monazite and REE-fluorocarbonates, fooling us to think that the mineralisation is hydrothermal, and the REE were deposited from fluids. Not really - the fluids simply caused

local redistribution of the minerals. This is why in more cases the mineralisation is confined to the ferrocarnatite cores of complexes, as these ferrocarnatites represent the last stages of carbonatite magma evolution. Had there been significant hydrothermal mobility on a large spatial scale, the mineralisation would be elsewhere. What I think is nice in your paper is that you show that in alkali carbonate fluids, HREE are more soluble than LREE. This is very similar to what we say in our SciAdv paper, where we show that LREE end up in the more insoluble phases, and HREE remain soluble. Thus, the HREE can migrate outwards to fenites, whereas LREE stay within the carbonatite, which is what we see in nature. I think this needs to be emphasised more in your paper. My fear is that people will see your paper and cite it without reading as an excuse to say "REE are mobile!" without paying too much attention to the nuances that you need alkali+F+CO₃ fluids, and the HREE are more soluble. This has happened before with the Migdisov papers, so try to be as clear as possible with it. I even suggest making a cartoon figure of the process you envision in nature based on your own study, just to be as clear as possible and prevent misinterpretation of your excellent work.

Note - I am not too familiar with the technicalities of the method (in-situ XANES, computational structure modelling, etc). I am writing this paper with the assumption that what the authors did is correct. Most of my comments are on the interpretations, the wider implications and how this might apply to natural systems.

Specific comments below. Apologies if they look a bit critical in some places - I love your paper :)

All the best and looking forward to seeing a revised version of the manuscript,
Michael Anenburg
Australian National University
michael.anenburg@anu.edu.au

(I am happy for the authors to contact me directly regarding specific issues).

line 13: "Rare earth elements", not "Rare Earth Elements".

line 14: Although not officially mandated, the people at IMA strongly recommend using "fluorcarbonate" and not "fluorocarbonate". I tend to agree with them. Here and elsewhere.

line 27: Although I'm sympathetic to statements like this, it is not actually correct (I'm also guilty of this). The greatest challenge nowadays is getting the REE out of the ore, not finding more REE ore.

line 32: Note that most REE deposits in carbonatites do not result from hydrothermal remobilisation, but rather from hydrothermal redistribution. That is, the REE are where they are now not because they were transported there by a hydrothermal fluid which is distinct from the carbonatite magma. They are there because an alkaline evolved carbonatite magma (or more like a magma-fluid hybrid thingy) concentrated the REE to the last stages of carbonatite evolution, and then deposited them initially as alkali REE carbonates like burbankite. Later less-alkaline hydrothermal fluids dissolved the burbankite (and others), and the REE were redistributed in-situ to the minerals we see now: the REE-fluorcarbonates. So yes, the "REE concentrations hosted in secondary minerals formed through hydrothermal alteration" as you say. But, the REE would still be there even without the hydrothermal alteration. The issue of remobilisation vs redistribution is discussed in the Cangelosi et al (2020a) paper that you cite, and also in Anenburg et al (2018).

line 39: bastnäsite

line 41: Goes back to my previous comment. Most of the fluid inclusions currently observed in carbonatites record the late redistribution stage. The mineralisation was already there before that. These fluids do sometimes contain CO₂, but its role is secondary as the minerals have already been deposited. There is "mobilisation" on any substantial scale by the fluids recorded in the fluid inclusions. The statement regarding the pH=3 fluid is, I am assuming, from the Trofanenko paper. Do not think that this was their modelling choice, and does not mean that pH=3 fluids have actually been observed. Many of us in the carbonatite community

have strong issues with this paper. Maybe one of the most obvious is how to get even get to have acidic fluids to begin with in a calcite-dominated system. Carbonatites simply do not exsolve anything acidic.. I strongly recommend reading the recent review by Walter et al (2021) to get a better idea of how fluids behave like in carbonatite systems.

line 43: They are "believed" to do so only by certain people. Again, fluids coming off carbonatites are not acidic to begin with, and they already carry their own load of phosphate and fluoride. REE phosphates and fluorides are very insoluble. Our own model (the SciAdv paper which you are familiar with) puts the finger at alkalis, and I think belongs here in the introduction?

line 53: "carbonatitic". And not a single one of those papers claims that they have pH=3.

line 101: Our fluids never "exsolved" per-se from the carbonatite melt. The carbonatite melt evolved into it, without phase separation. And we went down to 200C, not 400.

line 106: True! And that's what I was really hoping someone one study that, and then I got your paper. Well done :)

line 124: You haven't called Fig 1 yet.

line 140: It's like winning the jackpot, right?

line 148: F-free but carbonate bearing? Ie are those the "Na₂CO₃ - NaF" experiments from (iii) above?

line 149: Double periods.

fig 4: I'm guessing that red is O, violet is Gd, white is H and brown is C, but please label this.

line 297: No, not most genetic models. This entire acidic transport stems from the reason that people need REE to move somehow, especially in carbonatites, and the Migdisov & Willy-Jones paper provide this mechanism. Their papers are great and correct, but very often misinterpreted by the community. Upon close reading, it becomes obvious that they are relevant for acidic systems, and REE are immobilised in carbonatitic conditions. This doesn't stop many people from citing the papers as evidence for hydrothermal mobility, even without actually reading the papers. Note that not a single carbonatite paper (as far as I know) actually found convincing evidence for the presence of acidic fluids in a carbonatitic system! Apologies for the rant, it's not about your paper, but mostly about the lack of critical thinking in the community. I do, however, would appreciate if you could find a way to incorporate this into the paper. Something like "While most genetic models..no one has actually found these fluids in inclusions...".

line 302: Strange Lake is not a carbonatite! Your entire introduction was mostly about carbonatites. Make it very clear here that it's a silicate system.

line 315: This will never happen in a carbonatite - the pH of the system is rock buffered by the presence of carbonate minerals.

line 329: Maybe my comment for line 297 is more/also appropriate here.

line 337: Note that this very saline multicomponent fluids are most likely formed via protracted fractionation of the carbonatite magma, and not by exsolution. Essentially, these fluids are the carbonatite magma.

line 351: Ta? Carbonatites are ores for Nb, not Ta.

line 355: Not exsolved! Evolved continuously from the high-T carbonatite magma.

line 357: In the low thousands of ppm.. "close to wt%" is probably an overstatement :)

line 359: We quenched from 400, not to 200. Cooled down slowly to 200 (not 400), then quench.

line 363: Not quite. We said that solubility is strongly enhanced when alkalis are around. Clearly, there's something to charge balance the cations. We did not know what it is, and we speculated it is probably some kind of carbonate fluoride chloride complex. Maybe. In other words, we did reject carbonate, but we rejected only carbonate.

line 365: This misrepresents what we say. The silicate reaction releases CO₂. The question is what does this CO₂ end up as, and we showed that it ends up as CO₂ gas which is evident by the huge gas cavity in the experiment, and the lack of any quench-carbonate phases. The reason we think it did not end up as carbonate (and I think we touched upon it in the paper) is because the carbonate will have to be charge balanced by an anion, and none were available. The presence of carbonates (calcite, dolomite, etc) buffered pH so HCO₃⁻ complexes could not form, it could not be acidic enough. Thus the CO₂ just remained as CO₂ and didn't do much to the REE. This leads me to speculate about the differences. Here's an idea: When you have a system with insoluble carbonates (eg calcite) then there's no sufficient carbonate in the fluid. Once you have soluble carbonates (Na₂CO₃, K₂CO₃), and the pH is high, then you have everything in solution. The alkalis can give away their carbonate and pick up OH⁻ for charge balance, and the now available carbonate solubilises the REE. And this can only happen if pH is high, and alkalis are there. So we suggested alkali complexing, but reading your paper maybe a better idea is that the alkalis work indirectly by allowing carbonate to remain available for REE?

line 370: See above - it's probably not the amount of CO₂. We had loads of CO₂ available. I think that the issue was that it remained as CO₂, and did not dissolve in the H₂O as carbonate.

line 375: Back to exsolution. At ~500 you still have carbonatite magma which will strongly partition any REE that are around. Hydrothermal fluids, if present at this stage, are pretty much irrelevant for REE mobility. This is the stage where minerals like burbankite (and less commonly, primary bastnasite and monazite) start to crystallise, which then deplete the magma of the LREE. HREE remain in the magma, which at this stage of ~400 evolves (without exsolution) into something more like a hydrous brine, which by itself is, as you showed, very capable to keep HREE in solution.

line 378: By now you probably know what is my opinion regarding acidification (ie that it does not happen), but feel free to suggest a mechanism by which acidification happens if that is what you wish.

line 380: This becomes convoluted - you say that you start alkaline, become acidic, and then you need it to become alkaline again? In my opinion just drop the acidic part in between.

line 429: "compositions"

Also check out Horton et al: <https://doi.org/10.1029/2020GC009472> a new paper which talks a lot about fluids in carbonatites - might be useful?

Walter, B.F., Giebel, R.J., Steele-MacInnis, M., Marks, M.A.W., Kolb, J., and Markl, G., 2021, Fluids associated with carbonatitic magmatism: A critical review and implications for carbonatite magma ascent: *Earth-Science Reviews*, doi:10.1016/j.earscirev.2021.103509.

Anenburg, M., Burnham, A.D., and Mavrogenes, J.A., 2018, REE redistribution textures in altered fluorapatite: Symplectites, veins and phosphate-silicate-carbonate assemblages from the Nolans Bore P-REE-Th deposit, NT, Australia: *The Canadian Mineralogist*, v. 56, p. 331–354, doi:10.3749/canmin.1700038.

Reviewer #3 (Remarks to the Author):

This is a really interesting new set of results and I enjoyed reading the paper. However, I'm not an expert on the approaches used here, so my comments come more from a general awareness of ongoing research into REE mineral systems - I hope they are useful.

- Line 32-34: It's worth noting here that hydrothermal remobilisation is not always favourable, e.g. see Van de Ven et al 2019 <https://doi.org/10.3390/min9070422>
- Lines 37-45: There's some very recent work that could be cited here, e.g. Anenburg et al 2020 <https://advances.sciencemag.org/content/6/41/eabb6570> and Walter et al. 2021 <https://doi.org/10.1016/j.earscirev.2021.103509> - and a major point made in these papers is the importance of the alkali elements, which would be worth mentioning in your introduction. As I read on I see that you do refer to some of this work, but it might be worth highlighting earlier.
- Introduction, general: This is a very good introduction but it does focus very much on the theoretical/experimental without making a strong link to real geological situations. A particularly important point is that fenite haloes are very common around carbonatites, and these provide the real-world basis for study of alkaline fluids in REE transport. You might like to cite Elliott et al 2018 <https://doi.org/10.1016/j.oregeorev.2017.12.003>
- Line 116-119: What was the basis for focusing on Na as the main alkali element rather than K? It would be good to see a brief statement about this, either in the text or the methods section.
- Line 138: This seems to match with the experimental work of Song et al 2015 <https://link.springer.com/article/10.1007/s00410-015-1217-5>
- Line 159: The 'Goldilocks effect' mentioned here made me think of the higher-temperature work of Zineb Nabyl on immiscibility, which also showed that REE partitioning into carbonatites occurs in a particular compositional 'window' <https://doi.org/10.1016/j.gca.2020.04.008>. It's a different part of the system, so this is really just a comment of interest rather than an expectation that you change anything, but it all links in demonstrating the complex controls on REE solubility.
- Line 189-263: I am absolutely no expert on EXAFS analysis so can't really comment on this except to say that it is clearly written and makes sense for the non-expert! My only question would be whether, since it was only possible to study Gd and Yb, this section should refer to HREE rather than REE? Do you think the generalised structural parameters established here are also appropriate for the LREE?
- Line 266: It would be useful if you defined what you meant by high-T here.
- Line 292: Note spelling of bastnaesite
- Line 298-299: Please cite the studies referred to here
- Line 308: There's no such thing as a pegmatite melt! Alkali granitic melts might be the best term here.
- Line 302-328: This is quite a long review of work on Strange Lake, which although very well studied, is only one example. You might want to look at, for example, the work of Bernard et al 2020 <https://link.springer.com/article/10.1007/s00410-020-01723-y> for other examples.
- Line 321: Delete 'cousin'
- Line 332-336: What you've written here is accurate, but it implies the magmatic-hydrothermal transition has only been fairly recently recognised as being important in REE mineralisation in carbonatites, which isn't really correct. I'd like to see this section reworded to emphasise that there has been a lot of research on the magmatic-hydrothermal transition in these systems, maybe with a few key examples.
- Line 363-372: But your experiments and those of Anenburg et al covered different temperature ranges, so there is a need to be cautious about saying they contradict each other?
- Discussion, general: Overall, because your work focuses on the lower-temperature end of the system, I think this paper would benefit from placing the different processes and previous research into a clearer temperature framework. What do you consider is the temperature range of the magmatic-hydrothermal transition, for example? Fenitisation by alkaline fluids is generally considered to occur at >400°C (Elliott et al 2018), so how does this fit with your results? I suggest that you could shorten the summary of previous work on Strange Lake, to give space for a few lines summarising the broad evolution of these systems with temperature and where your new results fit in.

K M Goodenough April 2021

Dear Referees,

Please find attached the revised version of our manuscript entitled ‘Carbonate complexation enhances hydrothermal transport of Rare earth elements in alkaline fluids’.

On a general note, we would like to thank the editor and the three reviewers for their very constructive comments and suggestions, which, we believe, enabled us to significantly improve the original manuscript.

Based on the comments of Reviewer 1, we now provide additional details about the experimental and analytical methods. Especially, a new sub-section entitled ‘*High-temperature in-situ XAS: an experimental window into the hydrothermal behavior of REE*’ has been added at the beginning of the Result section (**lines 148-166**) to underline the advantages of in-situ XAS to study metals in hydrothermal fluids. Additional details and clarifications about P-T conditions, fluid compositions and the potential precipitation of secondary phases at high P-T conditions can be found throughout the Result and Discussion sections (**lines 162-166, 174-179, 343-346 or 366-369**).

While our original aim was to underline implications of REE-carbonate complexation for REE hydrothermal transport at large, we understand the reviewers’ call for a better explanation about how our experiments may relate to natural systems and therefore rewrote significant parts of the Introduction and Discussion:

- More details are now given in the Introduction about magmatic and hydrothermal concentration processes (**lines 32-44**), the impact of hydrothermal mobilization in different locations in term of LREE/HREE enrichment (**lines 48-67**) and the applications and limits of the acidic model to describe natural enrichments (**lines 75-108**).
- The evolution of the composition of ‘carbonatitic’ fluids from the magmatic-hydrothermal transition to later hydrothermal stages is also explicitly presented in the Discussion (**lines 380-432**). We further clarify that our experimental results are better adapted to describe REE transport mechanisms and solubility behaviour in ‘diluted’ hydrothermal fluids and provide key examples of the REE signatures that may be related to ‘melt-brines’ or ‘diluted’ fluids in natural systems (**lines 408-413 and 441-469**).
- Following on the comments of Reviewers 2 and 3, we also decided to refocus the discussion mostly towards REE mineralization in carbonatite and shorten the discussion on the role of alkaline fluids in peralkaline granitic deposits (e.g., Strange Lake) (**lines 475-488**).

All corrections have been marked as bold text in the manuscript (and in blue in the abstract) and answers to the Reviewer’s comments can be found below.

We hope you will receive these corrections positively.

Best regards,

Marion Louvel, on behalf of all authors

Reviewer #1 – Alex Gisy

This study presents a series of new hydrothermal experiments and in situ X-ray absorption spectroscopy for the solubility of REE in alkaline carbonate-fluoride-bearing aqueous fluids. The authors aim at relating their experimental results to the transport and mineralization process of REE in mineral deposits in natural systems. The manuscript compares previous experimental work done mainly at low pH and 300 C to their new results conducted at elevated temperature up to 500 C in alkaline fluids and try to draw conclusions for carbonatite and alkaline hydrothermal-magmatic systems.

Overall, I am excited that the solubility of REE was measured at elevated pressure and temperature, and that we are finally making advances on the speciation, not only in acidic environment, but in alkaline aqueous fluids, which to my opinion certainly play a key in natural REE mineral deposit ore formation. The authors seems to have identified the importance of carbonate complexes in alkaline fluids using spectroscopic analyses. This is a potentially important finding.

On the other hand the manuscript has some problems concerning some of the experimental data interpretations and their presentations in the text, and I encourage the authors to address these to target a broader audience and make it easier to follow. I added detailed comments in the annotated pdf. Here is a summary of the main points:

1) The introduction is somewhat difficult to follow and needs reorganization to better link the problem between our knowledge (or lack thereof) of thermodynamic properties of REE speciation and laboratory experiments vs. observations in natural REE deposits. There are some affirmative sentences without references to back up the ideas presented. Also a few of the references are not adequately cited including in the discussion and there are many typographical errors in the manuscript. I found the same problem in the discussion section where the authors discuss natural systems starting line 296.

Answer: We understand the reviewer's comments and rewrote most of the introduction to better introduce 1) the natural systems, 2) the acidic model and 3) available experimental data and how useful they are (or not) to describe the potential role of alkaline fluids in REE ore formation.

In line with the three reviewers' comments, we now provide more detailed examples of magmatic/hydrothermal concentration processes and LREE/HREE patterns at different locations (e.g., lines 35-67 or 80-95).

Comments about the discussion are addressed below.

2) In the results section, I suggest to be more explicit in the type of experiments that were used, perhaps even have a short paragraph that explains briefly the advantages of using in situ XAS in geosciences. There are also several places where temperature/pressure or condition of the experiments are not listed. I understand that the details go into the methods section, but it is difficult to separate some of the conditions. I also added some comments about this in the methods sections as the authors mention solubility vs. speciation experiments there but not in the text, and need to clarify the difference (analytical, experimental, or interpretation of spectra).

Answer: A new short subsection entitled '*High-temperature in-situ XAS: an experimental window into the hydrothermal behavior of REE*' was added at the beginning of the Results section to present the advantages of in-situ XAS and detail the P-T conditions of experiments (lines 148-166).

Other comments are addressed below.

3) The presentation of experimental results needs to be improved. I have not found any mention in the results explaining the rationale behind the initial setup of the fluid compositions used in the experiments, i.e. lines 116-119. Why did the authors exactly choose these compositions, did they try to isolate the control of REE complexation on REE solubility? The result section also clearly needs to state that the experiments measured the solubility of REE oxides, and explain possible secondary minerals that can form as a result of the presence of fluoride and carbonate in the fluid.

Answer: The rationale behind the investigated fluid composition was to study both the effect of pH and different ligands (OH-, F- or CO₃²⁻) on REE aqueous behavior. Details about the nature of starting materials, P-T conditions and controls on REE aqueous concentrations were thus added on lines 162-166 and 169-179.

Most of the reviewer's questions about secondary minerals are answered below. As no high T precipitates could be recovered, the potential nature of the precipitates behind the retrograde solubility of Gd and Yb and the lower La concentration in F-rich composition are mentioned in the discussion rather than the results (lines 343-346 and 366-369).

To me the solubility variations seen in Figure 2 can stem from several competing reactions including: i) REE complexation, ii) solubility of the REE solids that were used in the experiments, iii) the stability of secondary REE minerals. All 3 competing reactions depend on pH and temperature and ligand activity. The results only discuss total solubility of REE and the identification of possible REE carbonate aqueous complex in a few of the experiments. It would be beneficial for the reader to clarify that these processes are investigated and which experiment is controlled by which process.

Answer: The reviewer was right to mention the complexity of solubility behaviour under high P-T conditions. We are also well aware that addressing our concentration study as 'solubility' could be of concern to the solubility/thermodynamic community!

A paragraph mentioning the different controls (mineral solubility, stability and aqueous speciation) on REE aqueous concentrations has now been added (lines 162-166). Also, we tried to remove 'solubility' when possible and instead refer to 'concentration' or 'concentration trends' throughout the text.

4) Any possible carbonate or fluoride phases that could have formed in these experiments? How would that affect the measured spectroscopic signals? Why were these solids not measured after quenching the experimental fluids? Couldn't they yield valuable information on the experiments?

Answer: Regarding the nature of potential high *T* precipitates, which could indeed buffer the aqueous solubility, those could not be recovered after quench as they constitute a tiny amounts ($\ll 0.1\text{mg}$) of a very thin powder on the top and side of the lower piston rather than a chunk of solid.

We are however confident that such fine powder settled fast to the bottom of the sample cell and did not contaminate the measurements. Indeed, the presence of 'floaters' or of actual crystals on the walls of the cell is easily evidenced as fluorescence peaks recorded in transversal or vertical scans of the sample chamber. Furthermore, we found in previous measurements that REE₂O₃ or REEF₃ solids display characteristic post-edge features in the XANES spectra (Louvel et al., 2015).

Powders of REE₂(CO₃)₃.xH₂O were also analysed with XAS as reference materials. They also display a post-edge feature, shifted to slightly lower energies than in REE₂O₃ and REEF₃ (see Figure 3 in Louvel et al., 2015). None of these features were observed in the high T measurements for Gd and Yb in 0.7m Na₂CO₃ +-NaF solutions.

For La, solid LaCO₃F (synthetic) is for instance characterized by a post-edge shoulder centred at 5506eV. Despite the low quality of La XAS, it can be seen from the figure below that none of the 450°C spectra present such a feature, discarding the precipitation of large amounts of LaCO₃F under the beam at this temperature for all composition. Note that it doesn't mean that minute amounts of La solids did not accumulate at the top and on the side of the pistons, where they cannot be monitored in-situ. This is why we explain the decrease of La aqueous concentration observed in the 0.7m Na₂CO₃ + 0.6m Na₂CO₃ above 450°C as the result of the precipitation of La-fluorcarbonate.

5) I was confused in the discussion about which REE aqueous species was identified based on which evidence. Figure 4 and text in the results section indicates that a Gd hydroxyl-carbonate species was identified using ADF calculations. The experiments indicate also a possible role of F, although as said above, some experiments show the precipitation of fluoride/fluorocarbonates, which have not been identified. However, the discussion then mentioned (hydroxyl-)carbonate complexes (line 273), perhaps Sm hydroxy-fluoride complexes (lines 321-322), and finally a new (hydroxyl-, fluoride-)carbonate species (line 388). This needs to be better explained or clarified.

- We agree with the reviewer that our chosen speciation model was not as clear as possible in the original manuscript. To sum up, EXAFS analysis supports a structure involving 2 carbonate groups and 4 OH/H₂O groups in the first coordination shell and the presence of 2 additional REE atoms (plus some water) further away from the central REE atom. In the original manuscript, we argued that this could be accommodated either as a simple hydrated REE carbonate complex surrounded by two hydrated REE atoms or as a polynuclear hydroxyl-carbonate complex, the later being our structure of preference, in comparison to what is known for REE and actinides in alkaline fluids under room conditions. In the new version, we however found additional references (e.g., XXX) that also support the formation of hydrated carbonate complexed, based on uranyl complexes. ADF calculations are here used to illustrate those proposed structures, based on our EXAFS structural parameters. The all section as thus been reorganized to better present both models and how we used ADF calculations to illustrate our EXAFS findings (lines 296-314). I hope it makes more sense now. We also clearly state that our preferred structure remains that of hydroxyl-carbonate complex and not hydrated carbonate complex, because of the high pH (lines 311-314).

- Fluor-carbonate complexes could not be evidenced through EXAFS analysis, mostly due to the fact that O (ie., from OH groups) cannot be distinguished from F with this approach. Because of that, the potential formation of such complexes is not mentioned in the Result section, but only discussed in the Discussion, based on the higher Yb and La concentrations reported in the presence of F. The discussion of OH⁻ versus F and the ‘Goldilocks effect’ of F has also been reorganized to better distinguish what we know from EXAFS analysis (formation of Gd and Yb hydroxyl-carbonate complexes at 200°C) from our interpretation of the F effect on La and Yb aqueous concentrations as a potential evidence for mixed fluoride/carbonate complexation (lines 355-369).

5) A follow up on point 4), the results are not always conclusive, i.e. many of the experiments were below the limit of detection. A big question mark to the reader a bit less familiar with the spectroscopic methods could be: why are all the REE-OH experiments below the limit of detection, or are they not detectable at all at the measured conditions? Can the authors perhaps state whether any of their setup could be changed to better detect these species using the synchrotron, or is it possible to perhaps use higher REE concentrations in the starting fluids for these experiments?

Answer: All experiments involved dissolving a solid pellet of REE₂O₃ in a fluid. Those involving NaOH solutions resulted in extremely low solubility of the REE₂O₃, and thus REE aqueous concentrations below our detection limits of ~10ppm in fluorescence mode. This in turns results in rather ‘featureless’ spectra where only the absorption edge can be seen at best (e.g., Fig 1B) and makes the EXAFS analysis and the characterization of REE-OH complexes impossible.

As it is not possible to increase the solubility of the REE in the NaOH fluids, our set-up cannot be used to describe the REE-OH species. A solution to overcome this issue may be to reduce the thickness of the high-pressure windows so as to reduce the absorption of incoming X-rays and fluoresced photons. This is unfortunately not possible at the investigated pressures.

Overall, I see the effort the authors have undertaken to measure the solubility of REE at difficult conditions. While these are important new results, the points above need to be clarified to enable future research to build on these findings. I hope my comments will help improving the manuscript,

Alexander Gysi

Answer to line to line comments in the pdf:

Line 21: I assume the question is about why we here use ‘transport’ when our experiments were conducted in a closed system? I.e., we did not do reactive transport experiments in a flow-through device? A better term for the experiments could indeed be that carbonate complexes promote the simultaneous dissolution/solubility of REE, F and CO₃²⁻, but that would make it more of a technical sentence, when the aim of this short abstract is to already focus the reader’s attention on the implications of our study rather than the detailed experimental processes.

Lines 22, 24: I agree with the reviewer that the term ‘early’ may be confusing here, especially as ‘late magmatic fluids’ are also mentioned later on in the text to describe the same high T fluids (400 C and above). This is some mannerism of speech on my side and I thus tried to remove such mentions in most places.

Line 30-32: Reviewers 2 and 3 expressed the same concerns and also advised for a rewriting of the whole section describing natural occurrences. A modified text presenting magmatic and hydrothermal processes, the importance of the magmatic-hydrothermal transition and the natural constraints on LREE/HREE concentration processes can be found from lines 32 to 67 in the new manuscript.

Line 41: The same concern was expressed by reviewer 2. pH constraints are in general scarce and this value only referenced the work of Trofanenko et al. As explained above, the whole section has been changed to better present the natural occurrences, giving more examples for different deposits and further focusing on carbonatites and fenitization, as most of our results are for carbonate-rich fluids (as solubility was bdl in most carbonate-free systems).

Line 139: Regarding the distinction between LREE and HREE we here follow the EU definition (for instance mentioned in Goodenough et al. 2016), where LREE include La to Sm and HREE Eu to Lu, +Y. This separation makes sense when comparing the solubility trends of Gd and Yb (retrograde solubility) to the unexpected prograde solubility of La.

Based on our complementary studies on REE speciation in Cl-rich acidic fluids (unpublished), the actual separation in term of aqueous behaviour may be a bit more complicated between LREE (La, Nd and Sm – all elements have an hydration shell composed of 9 H₂O and tend to form similar chloride complexes at similar T), MREE (Eu and Gd – which kind of have an intermediary behaviour) and HREE (Er and Yb in our case – which only have 8 water molecules when fully hydrated and tend to form chloride complexes with less chlorine atoms than LREE). Y is another story, with a behaviour that evolves from HREE-like at low T to LREE-like at T > 300-400C (Guan et al., 2020).

Line 143-145: As mentioned above, we did not recover significant amounts of newly formed minerals. I would also expect that if La-(fluoro)carbonate had formed at high T in the experiments with 0.3m NaF, we would first have observed the dissolution of our La₂O₃, ie., an increase of La concentration in solution, at temperature of 100-200C. We did not observe any. However, we believe that increasing F concentration may favour the precipitation of fluorocarbonate at T>400C, as discussed on lines 362-369 (the ‘Goldilocks effect’).

Lines 155-160: The calculated pH is ~9.5 at 400C and 80MPa.

We discuss the reasons for the Goldilocks effect later on in the paper (lines 362-369), suggesting that too much F may favour the precipitation of REE-fluorocarbonates whose solubility would then control REE aqueous concentration. As mentioned above, we unfortunately could not recover enough quenched material to make sure that such fluorocarbonate formed and this hypothesis hence remains speculative. More experiments are still needed!

Line 162-166: The reviewer asked if the comparison to Yb solubility in HCl solution could instead be presented as a discussion of acidic vs. alkaline transport of REE later on in the manuscript. We understand this point of view, but working around the short format of Nature Communications, we chose to keep our discussion focused on the ‘Goldilocks’ effect of F and the potential role of alkaline fluids in natural systems and thus kept the comparison to Yb in HCl as a short sentence in the result section (lines 259-263). We here provide a description of (new) complexes that may favour the alkaline transport of REE, but I believe that further solubility experiments are still needed to start a more comprehensive discussion of acidic vs. alkaline transport of REE.

Line 192: We believe using REE transport makes it more understandable to the non-specialist reader and would hence like to keep this here.

The likeliness that we ‘missed’ REE-hydroxyl complexation due to technical limitation, is very low. REE-hydroxyls would be probed as well, if they enabled aqueous concentrations above our detection limit (>10ppm, but above 500-1000ppm for actual EXAFS analysis). The potential precipitation of nanoparticles or of a coating of REE minerals on the cell walls has been explained above.

Line 198-200: Details of the differences observed between Yb in CO₃²⁻+F solutions and 0.1mHCl were added to the sentence now on lines 259-263. The spectra in 0.1mHCl is unpublished as such, but it was collected during the same set of experiments, ie., at 200°C and 80 MPa. Detailed information and EXAFS analysis for this spectra are not presented here as we did not want to compare acidic and alkaline speciation but rather highlight that alkaline fluids may be important in the ore-formation through the description of solubility trends and REE-carbonate complexes. However, we acknowledge that EXAFS

analysis, which will be presented in a different publication compiling all our in-situ speciation analyses on REE (summary of more than 8 weeks of synchrotron experiments, presenting data on La, Nd, Sm, Gd, Er, Yb and Y in Cl and SO₄-rich fluids with varying acidic pH), suggest similar structure as reported for other Cl-bearing solutions (0.35m HCl) in Louvel et al., 2015 (ie., Yb³⁺ hydrated by 8H₂O at 200°C and 80MPa)

Line 211: The studies are cited at the end of the paragraph (now lines 281 and 284).

Line 272: The solids are now mentioned in the result section.

Line 273-275: I guess the question is how can we compare our experimental composition to natural fluid composition? This is now addressed in more details in the discussion, where we present petrological and fluid inclusion constraints on the composition of carbonatitic fluids at the magmatic-hydrothermal transition and during later purely hydrothermal stages (lines 392-426 and 444-469) and further state that our experimental results are better used to describe REE transport in the more diluted fluids (not in ‘melt-brines’ – lines 432-446).

Line 285: No, it is almost impossible to distinguish O from F as nearest neighbour with XAS, due to their similar ionic radii and interatomic distances. We therefore modified this sentence to mention that *‘Whether this solubility increase is favoured by combined fluoride (as REEF²⁺, REEF₂⁺ or hydroxyl-fluorides REEF_x(OH)_y) and hydroxyl-carbonate complexation or the substitution of F⁻ for OH⁻ in the [REE₃(CO₃)₂(OH)₄]⁺ polynuclear complexes however remains to be constrained. Unfortunately, similarities between O and F ionic radii and interatomic distances render discrimination between hydroxyl and fluorine neighbors extremely difficult with XAS [60]’* (lines 355-361).

Line 289: No, what we meant is that the large complexes we report in the fluids (which are not precipitates as explained above) may be precursors to mineral formation.

Lines 300, 306, 314-316, 328 : As the discussion has been redirected mostly towards carbonatitic systems, references to ore-forming processes and the fluid composition in peralkaline syenites and granites are now found in the Introduction (lines 62-67 and 80-89) and at the end of the discussion, as a call for new experiments (lines 475-492).

Line 332-336: This sentence has been removed from the modified discussion.

Lines 376-378 and 380-382: As mentioned above, the discussion has been significantly modified to account for the three reviewers’ advice to 1) better link our experimental compositions to natural systems and 2) better explain the evolution of fluid composition from the early magmatic-hydrothermal stages to later purely hydrothermal ones.

Line 467: Corrections were made according to the reviewer’s request.

Line 468: Yes, the experiments are the same, only the data analysis provides different information (in-situ concentrations or speciation). The corresponding subtitles were hence changed to match the reviewer’s advice (*Estimation of REE aqueous concentrations from XAS spectra and Speciation analyses*).

Figure 1: We believe it is useful to show the slight increase in Sm in NaOH+NaF fluids as we mention it in the result part (lines 191-195). However, we agree with the reviewer that those are only few data, just above our detection limit and hence decided not to discuss more REE in NaOH fluids. The need to cut the discussion of peralkaline granites was also suggested by the two other reviewers who thought our data were of better use to discuss carbonatitic systems.

Regarding the potential role of Na-OH binding in preventing REE-OH complexation, this is very likely. However, more data would be needed to discuss that, maybe using other technics that better enable the quantification of low concentrations (< or around 10 ppm) of REE.

Figure 2: Analyzing blank solution would result in having no absorption edge and hence no eH value. To test the effect of density, we present data for other solutions with known REE concentration (reference solutions) showing the evolution of eH as a function of decreasing density in the Supplementary Materials (Figure S2). We show that this effect is negligible to fluid densities of 0.7-0.8 g.cm⁻³ which correspond to temperature of 300-400°C. Above 400°C, this effect is difficult to estimate as even acidic solutions with initially low amounts of REE (200ppm) tend to lose REE to below detection limit, following typical retrograde solubility.

The presented experiments kind of underline our in-situ limits, at least with the hydrothermal autoclave. In the future, experiments conducted at higher pressure in diamond-anvil cells could place further constraints on the solubility of REE in alkaline fluids to T > 500°C, as higher P-T conditions should enable higher REE solubility.

Figure 4 : The two proposed structures are now presented in the Figure. All atoms as labelled.

Reviewer #2 – Michael Anenburg:

Dear editors and authors,

I love the paper by Louvel et al because it demonstrates what we inferred indirectly from our own experiments (Anenburg et al 2020): that alkalis in alkaline systems make REE (and especially HREE!) very soluble. I also love the paper because it shows rather convincingly that we were wrong about the mechanism. It is not directly complexing by alkalis, but rather (as I understand) that alkalis make carbonate available for the REE. It's always interesting when people find you are wrong..!

In terms of novelty and significance, I would definitely recommend this paper be published in NComms. The more nails we can put in the coffin of "acidic mobility of REE", the better.

That said, I still have some issues with the paper that I believe need to be addressed before acceptance.

1. The paper attempts to enhance the significance of their finding that alkaline fluids are responsible by emphasising the importance that others are giving to the acidic model. The issue is that no one actually found evidence for acidic fluids in their carbonatites, they just assume that this is the case, based on the (correct but misinterpreted) works of Migdisov and Willy-Jones. This current paper is already important without repeating those misinterpretations and fallacies made by others. As someone who is deeply involved in the carbonatite community, I would say that people don't support the acidic model because they actually see evidence for acidic fluids in their rocks, they cite it because they see hydrothermal mineralisation in their rocks and they need to cite something that justifies it (often without actually reading or understanding the papers they are citing!).

Answer: We thank the reviewer for this comment and added several sentences to the introduction to clarify the opposition between acidic and alkaline models and the fact that the former may not be at all adapted to carbonatite-related systems.

Especially, the acidic model, what it is based on (fluid inclusion studies and experiments), and what processes it may be applied to are now detailed from lines 68 to 95. Why this acidic model may not be adapted to carbonatite systems is further stressed from lines 96 to 108.

2. The magma-hydrothermal fluid dichotomy is inherited from the science of silica magma systems. Things aren't so clearly separated in carbonatite systems, where very often there is a continuous transition (without exsolution or phase separation) from carbonatite magma to saline brines, both of which can carry REE. You show retrograde solubility, that the REE contents drop at high temperature. But this is probably not too relevant to natural systems, where at those high temperatures the carbonatite magma still exists, and acts as the REE carrier.

Answer: It is obvious from the three reviewers' comments that there was a misunderstanding about how our investigated fluid composition (0.7m Na₂CO₃ +-NaF) may relate to natural (carbonatitic) systems, especially in the frame of the 'magmatic-hydrothermal' transition.

- We tried to clarify this by giving a more detailed presentation of how fluid composition may evolve in carbonatitic systems from the magmatic-hydrothermal transition to later hydrothermal stages (lines 392-432 and 441-469).
- We also insist that our experiments conducted in 'diluted' fluids (<10wt% solutes) may not be used to explain REE scavenging at the magmatic-hydrothermal transition, which both involves highly-concentrated 'melt-brines' and overall higher temperatures than the ones investigated here (T_{max} = 500C for La, but 300C for Gd and Yb) (lines 432-440).
- Instead, we suggest that REE-carbonate complexation may play a role in the later hydrothermal stages and further suggest that LREE/HREE fractionation may be a good mean to distinguish between REE-CO₃ and REE-SO₄ role in REE hydrothermal mobilization (lines 441-469).

3. Following on the above, there needs to be a clear separation in the text between silicate systems and carbonatite systems. Those are two completely different chemical environments.

Answer: This comment is in line with those of reviewers 1 and 3.

As our data are mostly applicable to carbonate-rich systems, ie., carbonatite-related deposits, we modified the discussion to mostly focus on these. The potential of alkaline transport in peralkaline silicate systems is still mentioned, but only as an opening and a call for future experiments (lines 475-488). The introduction was also modified to better distinguish silicate and carbonatite systems and how the acidic model may better apply to peralkaline granites but is highly questionable for carbonatites (lines 96-108).

4. In carbonatites, the REE mineralisation is often already spatially locked in place in the primary mineralisation from the carbonatite itself. However, those are initially assemblages of alkali minerals like burbankite and carbocearnite, which are very soluble once the Na is removed from the system in later fluids. So the minerals recrystallise in-situ to monazite and REE-fluorocarbonates, fooling us to think that the mineralisation is hydrothermal, and the REE were deposited from fluids.

Not really - the fluids simply caused local redistribution of the minerals. This is why in more cases the mineralisation is confined to the ferrocarnatite cores of complexes, as these ferrocarnatites represent the last stages of carbonatite magma evolution. Had there been significant hydrothermal mobility on a large spatial scale, the mineralisation would be elsewhere. What I think is nice in your paper is that you show that in alkali carbonate fluids, HREE are more soluble than LREE. This is very similar to what we say in our SciAdv paper, where we show that LREE end up in the more insoluble phases, and HREE remain soluble. Thus, the HREE can migrate outwards to fenites, whereas LREE stay within the carbonatite, which is what we see in nature. **I think this needs to be emphasised more in your paper.** My fear is that people will see your paper and cite it without reading as an excuse to say "REE are mobile!" without paying too much attention to the nuances that you need alkali-F+CO₃ fluids, and the HREE are more soluble. This has happened before with the Migdisov papers, so try to be as clear as possible with it. I even suggest making a cartoon figure of the process you envision in nature based on your own study, just to be as clear as possible and prevent misinterpretation of your excellent work.

Answer: This is a very complex issue.. As I mentioned above, I think there was a misunderstanding about how our 'diluted' experimental fluid composition can be compared to fluid composition in carbonatite-related deposits.

To clarify this, I first tried to better explain the role of melt-brines, especially in pre-concentrating REE in minerals as burbankite, as you presented in your Science Advances paper (lines 426-432). It is very obvious from the fluid inclusion record that this is an important step in the ore-forming process.

Then, I focus on how later hydrothermal fluids (diluted by some extent, either through precipitation of solutes or mixing with meteoritic fluids) may end up mobilizing LREE as carbonate complexes at T~300-450C, as suggested by our experiments. I believe that both focused vein formation and some 'in-situ' redistribution by replacement of carbonatite magmatic phases may attest to that (examples provided on lines 444-465). I hope you will agree with that.

The discussion of our observed LREE/HREE fractionation patterns versus the one you reported in the Science Advance paper is, I believe, a bit more difficult to introduce, especially as both our experiments are 'preliminary reports'.

First of all, I am not sure if and how solute-rich melt-brines may survive to low T conditions, below 300C. I believe you did more experiments than you have yet published, so would you have some input on that? Like, did you conduct similar experiments with quench at higher T (400-500C)? As the autoclave is not well-suited for the study of very dense fluids, investigating the nature of REE complexes in melt-brines in diamond-anvil cells is definitely on my to do list for the next years!

Secondly, I failed to find low T examples where HREE enrichment was due to their preferential transport, and not to preferential leaching of the LREE. Therefore, I believe that while the high solubility we report for Gd and Yb at 100-200C is interesting, it may not have much applications to natural systems. Maybe more to processing.

Note - I am not too familiar with the technicalities of the method (in-situ XANES, computational structure modelling, etc). I am writing this paper with the assumption that what the authors did is correct. Most of my comments are on the interpretations, the wider implications and how this might apply to natural systems.

Specific comments below. Apologies if they look a bit critical in some places - I love your paper :)

All the best and looking forward to seeing a revised version of the manuscript,

Michael Anenburg
Australian National University
michael.anenburg@anu.edu.au

(I am happy for the authors to contact me directly regarding specific issues).

line 13: "Rare earth elements", not "Rare Earth Elements".
Done

line 14: Although not officially mandated, the people at IMA strongly recommend using "fluorcarbonate" and not "fluorocarbonate". I tend to agree with them. Here and elsewhere.
Done

line 27: Although I'm sympathetic to statements like this, it is not actually correct (I'm also guilty of this). The greatest challenge nowadays is getting the REE out of the ore, not finding more REE ore. We agree but chose to keep the sentence as an introduction to the topic. It is quite general, and we do not say this is the greatest challenge, but more acknowledge that part of the mining operation could be improved by a better understanding and modelling of ore-forming processes.

line 32: Note that most REE deposits in carbonatites do not result from hydrothermal remobilisation, but rather from hydrothermal redistribution. That is, the REE are where they are now not because they were transported there by a hydrothermal fluid which is distinct from the carbonatite magma. They are there because an alkaline evolved carbonatite magma (or more like a magma-fluid hybrid thingy) concentrated the REE to the last stages of carbonatite evolution, and then deposited them initially as alkali REE carbonates like burbankite. Later less-alkaline hydrothermal fluids dissolved the burbankite (and others), and the REE were redistributed in-situ to the minerals we see now: the REE-fluorocarbonates. So yes, the "REE concentrations hosted in secondary minerals formed through hydrothermal alteration" as you say. But, the REE would still be there even without the hydrothermal alteration. The issue of remobilisation vs redistribution is discussed in the Cangelosi et al (2020a) paper that you cite, and also in Anenburg et al (2018).

This comment, as well as some others below, are in line with comments from Reviewer 1 and 2 who advised for a better description of natural occurrences, alterations and the link between fluid composition/properties and REE mineralization. Consequently, we rewrote large parts of the Introduction and Discussion. I hope you find the description of REE in-situ redistribution or concentration in veins/fenites better (lines 392-432 and 444-469).

I would however like to challenge your view of hydrothermal remobilisation/redistribution terms. From my point of view, those words mean the same thing. The only difference I could see with redistribution is that this term must involve some extent of LREE/HREE fractionation (i.e, the Cambridge dictionary defines redistribution as the act of sharing something out differently from before). ‘Hydrothermal mobilization’ or ‘remobilization’ has also been used for decades both in the context of pegmatite enrichment (ie. Gysi et al., 2013), where one can as well assume that the magmatic fluids are direct products of the extreme crystallization, and in the context of fenitisation.

I didn’t find the Cangelosi et al (2020a) paper to discuss that issue as clearly as you suggest. They for instance mention both in the following paragraph:

‘While there is a qualitative match between the elements released from magmatic minerals and those taken up in stage 3 hydrothermal phases, the most altered samples show some enrichment, implying redistribution on a larger scale than the outcrops sampled (e.g. Fig. 3b). In particular, remobilisation gave rise to a greater local LREE enrichment in the most intensely altered calcite carbonatite (OKC19-2) reflecting abundant stage 3 parsite mineralisation (Figs 4k, 10).’

Therefore, I decided to mostly continue using the term ‘remobilization’, except when a redistribution of the LREE and HREE was actually evidenced.

line 39: bastnasite

Done

line 41: Goes back to my previous comment. Most of the fluid inclusions currently observed in carbonatites record the late redistribution stage. The mineralisation was already there before that. These fluids do sometimes contain CO₂, but its role is secondary as the minerals have already been deposited. There is "mobilisation" on any substantial scale by the fluids recorded in the fluid inclusions. The statement regarding the pH=3 fluid is, I am assuming, from the Trofanenko paper. Do note that this was their modelling choice, and does not mean that pH=3 fluids have actually been observed. Many of us in the carbonatite community have strong issues with this paper. Maybe one of the most obvious is how to even get to have acidic fluids to begin with in a calcite-dominated system. Carbonatites simply do not exsolve anything acidic. I strongly recommend reading the recent review by Walter et al (2021) to get a better idea of how fluids behave like in carbonatite systems.

As mentioned above, the different fluids and how our experiments relate to them are now precised, mostly in the Discussion.

In the Introduction, the reference to Trofanenko et al. has actually been removed after our reorganization. The reference to Walter was added. The fact that acidic fluid transport is highly questionable is first mentioned in the Introduction (lines 96-108). In the discussion, we further precise how REE-carbonate or REE-sulfate may play a role in different setting (ie., REE-sulfate may be stable to near-neutral/slightly basic conditions according to Migdisov’s model)(lines 455-469).

line 43: They are "believed" to do so only by certain people. Again, fluids coming off carbonatites are not acidic to begin with, and they already carry their own load of phosphate and fluoride. REE phosphates and fluorides are very insoluble. Our own model (the SciAdv paper which you are familiar with) puts the finger at alkalis, and I think belongs here in the introduction?

You will see that in the new version of the Introduction, the difficulty to apply the acidic model to carbonatitic systems and the need to transport together F and REE are both introduced based on petrology and then on experimental aspects. I hence chose to keep the mention to your paper with the experimental work (lines 108-130).

However the details of your model, and the fact that melt-brine fluids are probably critical to pre-concentrate REE in phases as burbankite (and hence constitute the first step in forming an economic concentration) are now clearly stated in the Discussion (lines 426-440). Based on the effect of alkalis you observed and my previous experience of complexation in ‘solute-rich’ fluids (silicate ones, where we observed large alkali-silicate cluster with zirconium – Louvel et al., 2013), I further suggest that large complexes, involving both carbonates and alkali could account for the high solubility of REE in melt-brines. We will have to conduct more in-situ XAS in HDAC to test that hypothesis (I hope in 2023)!

line 53: "carbonatitic". And not a single one of those papers claims that they have pH=3.

Mention to pH has been removed from the new Introduction.

line 101: Our fluids never "exsolved" per-se from the carbonatite melt. The carbonatite melt evolved into it, without phase separation. And we went down to 200C, not 400.

It's a missuse on my side. And sorry for the typo, I never meant to write 400C!

line 106: True! And that;s what I was really hoping someone one study that, and then I got your paper. Well done :)

Thanks.

line 124: You haven't called Fig 1 yet.

Done (now line 185).

line 148: F-free but carbonate bearing? Ie are those the "Na₂CO₃ - NaF" experiments from (iii) above? Yes. At the beginning of the section, Na₂CO₃ ±NaF are mentioned. This is because we investigated both 0.7m Na₂CO₃ (F-free) and 0.7mNa₂CO₃+some NaF.

In the light of further comments from Reviewer 1, we actually modified the result presentation so as to better present the rationale behind the investigated compositions (lines 169-179).

fig 4: I'm guessing that red is O, violet is Gd, white is H and brown is C, but please label this.

Done

line 297: No, not most genetic models. This entire acidic transport stems from the reason that people need REE to move somehow, especially in carbonatites, and the Migdisov & Willy-Jones paper provide this mechanism. Their papers are great and correct, but very often misinterpreted by the community. Upon close reading, it becomes obvious that they are relevant for acidic systems, and REE are immobilised in carbonatitic conditions. This doesn't stop many people from citing the papers as evidence for hydrothermal mobility, even without actually reading the papers. Note that not a single carbonatite paper (as far as I know) actually found convincing evidence for the presence of acidic fluids in a carbonatitic system! Apologies for the rant, it's not about your paper, but mostly about the lack of critical thinking in the community. I do, however, would appreciate if you could find a way to incorporate this into the paper. Something like "While most genetic models..no one has actually found these fluids in inclusions...".

As mentioned above, the Discussion was significantly modified after yours and the other reviewers' comments. Regarding a sentence that clearly state that some contributions may misuse the thermodynamic model of Migdisov, this is now stated on lines 380-386, with additional comments on the potential role of REE-sulfates on lines 465-469.

line 302: Strange Lake is not a carbonatite! Your entire introduction was mostly about carbonatites. Make it very clear here that it's a silicate system.

The discussion was changed accordingly to only mention Strange Lake (and other syenites/granites) as a 'call' for new experiments (lines 475-492). While alkaline fluids are expected in these systems, they may lack carbonates. Though the one experiment we conducted for Sm did show an increase in aqueous concentration in NaOH+NaF, we acknowledge this is yet too few to draw conclusions about REE transport in carbonate-free alkaline fluids.

line 315: This will never happen in a carbonatite - the pH of the system is rock buffered by the presence of carbonate minerals.

line 329: Maybe my comment for line 297 is more/also appropriate here.

line 337: Note that this very saline multicomponent fluids are most likely formed via protracted fractionation of the carbonatite magma, and not by exsolution. Essentially, these fluids are the carbonatite magma.

line 351: Ta? Carbonatites are ores for Nb, not Ta.

line 355: Not exsolved! Evolved continuously from the high-T carbonatite magma.

line 357: In the low thousands of ppm.. "close to wt%" is probably an overstatement :)

line 359: We quenched from 400, not to 200. Cooled down slowly to 200 (not 400), then quench.

All the above comments are not addressed as the Discussion was changed.

line 363: Not quite. We said that solubility is strongly enhanced when alkalis are around. Clearly, there's something to charge balance the cations. We did not know what it is, and we speculated it is probably some kind of carbonate fluoride chloride complex. Maybe. In other words, we did reject carbonate, but we rejected only carbonate.

line 365: This misrepresents what we say. The silicate reaction releases CO₂. The question is what does this CO₂ end up as, and we showed that it ends up as CO₂ gas which is evident by the huge gas cavity in the experiment, and the lack of any quench-carbonate phases. The reason we think it did not end up as carbonate (and I think we touched upon it in the paper) is because the carbonate will have to be charge balanced by an anion, and none were available. The presence of carbonates (calcite, dolomite, etc) buffered pH so HCO₃ complexes could not form, it could not be acidic enough. Thus the CO₂ just remained as CO₂ and didn't do much to the REE. This leads me to speculate about the differences. Here's an idea: When you have a system with insoluble carbonates (eg calcite) then there's no sufficient carbonate in the fluid. Once you have soluble carbonates (Na₂CO₃, K₂CO₃), and the pH is high, then you have everything in solution. The alkalis can give away their carbonate and pick up OH⁻ for charge balance, and the now available carbonate solubilises the REE. And this can only happen if pH is high, and alkalis are there. So we suggested alkali complexing, but reading your paper maybe a better idea is that the alkalis work indirectly by allowing carbonate to remain available for REE?

As mentioned above, one candidate could be large alkali-carbonate clusters...

line 375: Back to exsolution. At ~500 you still have carbonatite magma which will strongly partition any REE that are around. Hydrothermal fluids, if present at this stage, are pretty much irrelevant for REE mobility. This is the stage where minerals like burbankite (and less commonly, primary bastnasite and monazite) start to crystallise, which then deplete the magma of the LREE. HREE remain in the magma, which at this stage of ~400 evolves (without exsolution) into something more like a hydrous brine, which by itself is, as you showed, very capable to keep HREE in solution.

This has been addressed above.

line 429: "compositions"

Done

Also check out Horton et al a new paper which talks a lot about fluids in carbonatites - might be useful?

Walter, B.F., Giebel, R.J., Steele-MacInnis, M., Marks, M.A.W., Kolb, J., and Markl, G., 2021, Fluids associated with carbonatitic magmatism: A critical review and implications for carbonatite magma ascent: *Earth-Science Reviews*, doi:10.1016/j.earscirev.2021.103509.

Anenburg, M., Burnham, A.D., and Mavrogenes, J.A., 2018, REE redistribution textures in altered fluorapatite: Symplectites, veins and phosphate-silicate-carbonate assemblages from the Nolans Bore P-REE-Th deposit, NT, Australia: *The Canadian Mineralogist*, v. 56, p. 331-354, doi:10.3749/canmin.1700038.

Reviewer #3 – Kathryn Goodenough

This is a really interesting new set of results and I enjoyed reading the paper.

However, I'm not an expert on the approaches used here, so my comments come more from a general awareness of ongoing research into REE mineral systems - I hope they are useful.

Line 32-34: It's worth noting here that hydrothermal remobilisation is not always favourable, e.g. see Van de Ven et al 2019 <https://doi.org/10.3390/min9070422>

The fact that hydrothermal remobilization may lead to both reconcentration and dissemination of the REE is now mentioned both in the Introduction (lines 62-67) and at the end of discussion (lines 485-488).

Lines 37-45: There's some very recent work that could be cited here, e.g. Anenburg et al 2020 and Walter et al. 2021 <https://doi.org/10.1016/j.earscirev.2021.103509> - and a major point made in these papers is the importance of the alkali elements, which would be worth mentioning in your introduction. As I read on I see that you do refer to some of this work, but it might be worth highlighting earlier.

Following on all three reviewers' advices, we modified the introduction quite significantly to better introduce the magmatic and hydrothermal processes and how the 'acidic model', with REE transported as REE-Cl or REE-SO₄ complexes may not be adapted to model hydrothermal remobilization in carbonatite-related ore deposits. The mention to carbonatite/fenites and Walter's work can be found on lines 98-102. We kept the first mention to Anenburg et al. 2020 with the summary of available experiments for REE in alkali-rich fluids (lines 109-130).

Introduction, general: This is a very good introduction but it does focus very much on the theoretical/experimental without making a strong link to real geological situations. A particularly important point is that fenite haloes are very common around carbonatites, and these provide the real-world basis for study of alkaline fluids in REE transport. You might like to cite Elliott et al 2018.

As mentioned above, the Introduction has been significantly changed. A more detailed presentation of natural occurrences and the importance of fenites can now be found on lines 32-67 and 98-12. The detailed description of the evolution of fluid composition at the magmatic-hydrothermal transition and in later hydrothermal stages has been kept for the discussion (see below).

Line 116-119: What was the basis for focusing on Na as the main alkali element rather than K? It would be good to see a brief statement about this, either in the text or the methods section.

There was no strong basis for choosing Na over K. The 0.7m Na₂CO₃ concentration were chosen based on the previous high P-T work of Tsay et al. (2014), which provided evidences for enhanced HREE solubility in carbonate-rich fluids at 800°C and 2.6 GPa. Based on Anenburg et al. recent work, we acknowledge that investigating the effect of different (and combined) alkalis on REE solubility would be of interest and should be the scope of future solubility experiments. However, we do not expect the nature of REE complexes to be that different in the presence of both alkalis.

The reference to Tsay et al. (2014) was added to the introduction as a previous proof of REE solubility in Na₂CO₃-bearing fluids and to the method.

Line 138: This seems to match with the experimental work of Song et al 2015. <https://link.springer.com/article/10.1007/s00410-015-1217-5>

Song et al. indeed reported slightly higher D_{fluid/melt} for the REE in the presence of F. This was however in experiments where fluids were at equilibrium with a carbonatitic melt and hence probably carried some carbonates. Also, their data were collected at much higher temperature (800C). Thus, it is difficult to compare their results to our results on NaOH fluids at 200C.

We however decided to add this reference to others discussing the transport potential of fluids of carbonatitic origin in the introduction (Lines 114-122).

Line 159: The Goldilocks effect mentioned here made me think of the higher-temperature work of Zineb Nabyl on immiscibility, which also showed that REE partitioning into carbonatites occurs in a particular compositional window. It's a different part of the system, so this is really just a comment of interest rather than an expectation that you change anything, but it all links in demonstrating the complex controls on REE solubility.

We agree with you and really hope that both type of studies (melt-melt immiscibility and fluid solubility/speciation) will encourage others to place further constraints on REE behaviour from the early magmatic to the late hydrothermal concentration process. The reference to Nabyl et al. was however added to the introduction (line 37).

Line 189-263: I am absolutely no expert on EXAFS analysis so can't really comment on this except to say that it is clearly written and makes sense for the non-expert! My only question would be whether, since it was only possible to study Gd and Yb, this section should refer to HREE rather than REE? Do you think the generalised structural parameters established here are also appropriate for the LREE?

Thank you for this comment. I chose to only quickly mention that we believe a similar structure could explain the increase in LREE aqueous concentration at $T > 300\text{C}$ in the Discussion (Lines 348-351).

In details, small differences in the geometry (ie., bond distance, number of water molecules in the second shell) are probably to be expected, but we do not think that the amount of carbonate/fluoride present may enable completely different structures. This is for instance what we observe for LREE-Cl vs. HREE-Cl complexation, where LREE may bond to a slightly higher number of Cl under lower temperature than the HREE when high Cl concentration ($>20\text{ wt}\%$) are involved. Yet, under 'moderate' Cl concentration the dominant complexes are the same for LREE, HREE and Y (ie., REECl_2^+ at $T > 300\text{C}$) (Louvel et al., 2015; Guan et al., 2020; Guan et al. under review; and unpublished data).

One of my future goal is indeed to investigate LREE-carbonate complexation, but this will require to conduct in-situ XAS at the LREE K-edge (energy $> 35\text{keV}$). Such conditions are unfortunately not attainable at the BM30B beamline at ESRF where the autoclave is currently installed, but a similar experimental set-up is being implemented at the DESY synchrotron in Germany, on a beamline that enables higher energy. Another possibility will be to conduct diamond-anvil cell experiments, although those impose significantly higher pressure conditions ($>500\text{ MPa}$).

Line 266: It would be useful if you defined what you meant by high-T here.

Done

Line 292: Note spelling of bastnasite

Done

Line 298-299: Please cite the studies referred to here

This sentence has been removed.

Line 308: There's no such thing as a pegmatite melt! Alkali granitic melts might be the best term here. This sentence has also been removed following the reorganization of the Discussion.

Line 302-328: This is quite a long review of work on Strange Lake, which although very well studied, is only one example. You might want to look at, for example, the work of Bernard et al 2020 for other examples.

Following on yours and other reviewers' comments, this part has been removed and only kept as a 'call' for future experimental and fluid inclusion studies at the end of the discussion, together with references for Illimaussaq (lines 475-495). The Bernard et al., 2020 citation was added to the Introduction (lines 80-84).

Line 321: Delete cousin

Done

Line 332-336: What you've written here is accurate, but it implies the magmatic-hydrothermal transition has only been fairly recently recognised as being important in REE mineralisation in carbonatites, which isn't really correct. I'd like to see this section reworded to emphasise that there has been a lot of research on the magmatic-hydrothermal transition in these systems, maybe with a few key examples.

The entire section has been rewritten. A more detailed description of the magmatic-hydrothermal transition, the composition of early fluids and their role in ore-formation can now be found from lines 392 to 432. Several references were added to acknowledge the fact that different teams have been working on this subject for a long time (e.g., Rankin, Buhn et al.).

Most importantly, we now clearly explain that our experimental results may not be used to draw direct conclusions about the early solute-rich fluids ('melt-brines') but rather to discuss later hydrothermal reworking or vein formation by more diluted fluids (lines 432-469).

Line 363-372: But your experiments and those of Anenburg et al covered different temperature ranges, so there is a need to be cautious about saying they contradict each other?

This whole section was removed from the new Discussion. The results of Anenburg are mentioned to explain 'early' REE mobilization and formation of burbankite (lines 426-432).

Discussion, general: Overall, because your work focuses on the lower-temperature end of the system, I think this paper would benefit from placing the different processes and previous research into a clearer temperature framework. What do you consider is the temperature range of the magmatic-hydrothermal transition, for example? Fenitisation by alkaline fluids is generally considered to occur at 400C (Elliott et al 2018), so how does this fit with your results? I suggest that you could shorten the summary of previous work on Strange Lake, to give space for a few lines summarising the broad evolution of these systems with temperature and where your new results fit in.

Thank you for the advice. Reviewers 1 and 2 expressed similar concerns about the discussion of Strange Lake and peralkaline granite systems being too long, especially in the light of our few results on carbonate-free alkaline fluids (ie., one set of experiments showing increase of Sm solubility in NaOH+NaF fluids). Therefore, we decided to mostly cut out this part of the discussion and instead develop the discussion on carbonatite systems, as suggested.

As mentioned above, the entire discussion was significantly rewritten to better explain the different magmatic-hydrothermal and hydrothermal stages that may be at stake in carbonatite systems, and provide examples of places where REE-carbonate complexation in 'diluted' fluids may play an important role. I hope you like it!

REVIEWERS' COMMENTS

Reviewer #1 (Remarks to the Author):

Review manuscript entitled "Carbonate complexation enhances hydrothermal transport of rare earth elements in alkaline fluids"

This is the edited version of this manuscript after the first round of reviews. The authors did a great job addressing all my comments. I recommend publication of the manuscript and agree with most of their revisions and/or rebuttal letter. Below I have a couple of final remarks and sentence edits. I will leave it to the author's and editor's discretion to accept these.

Good work.

A. Gysi

abstract: add abbreviation for heavy (H)REE or light (L)REE

line: 15-19: this sentence is ambiguous, please rephrase. The only T the REE speciation is well established is <300-350 C, there are not much reliable thermodynamic data available above those temperatures. In the low temperature range, we still need to explore what could control speciation in alkaline fluids. We are currently working on REE hydroxyl complexes, their properties are mostly unknown. And then of course the carbonate complexes need to be studied...

Line 43: fluor-carbonates (bastnäsite, parisite,

Line 46: A key remaining question in these mineral deposits concerns the extent and conditions of...

Lines 49-56:

I suggest: Hydrothermal mineralization of REE in carbonatite systems generally results from the preferential mobilization of LREE over the HREE and the precipitation of strongly LREE-enriched phosphates and fluorcarbonates in veins and breccias. However, several carbonatite-related occurrences also present relative enrichments in the economically more valuable HREE in late-hydrothermal xenotime and/or fluorapatite

I am not sure about what is meant by this part, delete?: , but also more surprisingly in LREE-rich fluorcarbonates as parisite/synchysite

Lines 60-63: I do not understand well the sentence below. Calcite has commonly a preference for the LREE due to similar ionic radius with Ca. Also why the word "non-selective" if there is an enrichment of HREE over LREE, it would be selective.

A combination of primary source and magmatic enrichment processes (e.g., HREE concentration in magmatic calcite by fractional crystallization) followed by non-selective secondary hydrothermal remobilization of the REE [12,14].

Lines 63-69: here it would be appropriate to add Strange Lake (peralkaline granite/pegmatite).

It is one of the most extreme cases of hydrothermal REE and Zr mobilization and a world-class REE deposit. We have shown in the Gysi et al. (2016) paper in Econ Geol that there are almost no primary HREE minerals left, and that the deposit scale observations show that the HREE have been mobilized together with Zr. We also signal lithogeochemical vectors that show that F-metasomatism resulted in REE mobilization and enrichment at the deposit scale. Nechalacho is being mined now but most of the system is controlled by

magmatic mineralization, the system is more primitive than SL, and less hydrothermal veining and fluids (still worth mentioning though). Strange Lake also has hydrothermal fluorite breccia. This is just FYI, no need to add any of these details in the abstract.

Lines 70- 87: this is nicely explained!

Lines 98-100: My suggestion is to make the point further above that in peralkaline systems, the veins were mostly quartz and fluorite, which can be buffer fluids that remain at acidic conditions. In contrast, in carbonatites, the high alkali contents of the fluids (finitization), and the presence/abundance of calcite veins, suggest that the fluids could be buffered to alkaline conditions, and therefore the hydrothermal REE transport mechanisms may differ significantly to what we observed in the other natural systems.

My personal opinion, is that there is no need to create at this point a "camp" who favors an acidic or alkaline fluid model, because the chemistry of these fluids can also be dynamic. Whether the fluids are alkaline or acidic... I think it depends on the system, the initial fluids released from the magma, and their later interaction with minerals that can easily buffer the pH to higher values. The truth is we don't know well the chemistry of aqueous fluids in carbonatite systems, that is why experimental work can partly shed light on that, but we are far from being conclusive. The fluid inclusion work is still sparse to my knowledge, perhaps because workable fluid inclusions are harder to find.

For the example from China (lines93-97), I suppose the veins could be acidic if the calcite shows dissolution textures, would need to have a closer look at these studies. But if we precipitate calcite, then of course we could be near-neutral to alkaline.

I would reformulate and say that many field evidences like xxx suggest that the fluids in carbonatites may have an affinity for alkaline pH and therefore the mechanisms of REE transport/depositions may be quite different from what we know in other deposits and the acidic model.

Lines 102-104: please rephrase. Alkali-rich (Na, K) fluids is not necessarily equal to alkaline pH fluids... as you know these words have a different meaning. Carbonic acid is also a weak acid, but could buffer your pH to mildly acidic conditions.

Rest of intro is well written and nicely lays out the problem.

Line 254: add charge of fluoride.

Table 1: indicate what N, R, sigma etc. represent in table caption.

Lines 341-343: Why is a study from Chakhmouradian in elements cited here? I assume you mean Pourtier et al.?

I highly recommend removing "suggesting that currently available thermodynamics for hydroxide complexes may be used to such P-T conditions [1]". We showed in a couple of REE phosphate solubility experiments since 2015 that between 100 and 250 C, at least for the REEOH++ complex, that the data of Haas et al. does not predict well solubility data (see Gysi et al., 2018 GCA; Gysi and Harlov, 2021, Chem Geol).

Line 346: replace "alkalin transport" by transport in alkaline fluids.

Line 350: replace "alkalin transport" by LREE transport in alkaline fluids.

Lines 361-366: thanks for clarification!

Line 454: if you use "fluor" in carbonates, I suggest to also use this term for phosphates instead of "fluoro" to stay consistent. Same on line 480. I would use "fluorocarbonate" but since you already changed the other

parts to "fluorcarbonates" I suggest to correct here.

Reviewer #2 (Remarks to the Author):

Paper is mostly ok now.
Some minor clarity issues here and there - see attached file.

Reviewer #3 (Remarks to the Author):

I really enjoyed reading the rewritten introduction and discussion, and look forward to seeing the manuscript published. There are one or two minor spelling mistakes (e.g. Migdisov, line 72; bastnaesite, line 430) that should be corrected at proof stage.

Kathryn Goodenough

Answer to reviewers

Reviewer 1:

- Most vocabulary and rephrasing suggestions and typos have been corrected according to the reviewer's comments.
- Line 49-52: The reviewer asked the sentence to be rephrased as the speciation of REE is only well-constrained to 350C, and mostly to acidic conditions. The aim of this sentence is to underline in a broad way that while the role of high-temperature fluids has been well established from field studies, the mechanisms behind hydrothermal transport (including speciation) are not yet well-understood. The details of that can only be developed in the introduction as the abstract word number is limited to 150 words. Therefore, we chose not to change the sentence.
- Lines 81-86: We thank the reviewer for the suggestion, their proposed rephrasing is clearer than the original sentence.
- Lines 90-93: This was, as we understood it, the particular suggestion from the authors referenced for the Huanglongpu deposit in China: HREE would concentrate in magmatic calcite and hydrothermal transport, controlled by sulfate (hence 'non-selective'), would then remove both LREE and HREE, without further fractionation.
- Lines 97-99: The mention to Strange Lake has been added as suggested. The processes ongoing there are described later in the text (lines 114-124).
- Line 131-135: The abundance of calcite veins was also underlined as a main argument for not too acidic conditions by reviewer 2. I tried to follow Reviewer 1 advice to not exclude the potential role of acidic fluids by mentioning that 'at least part of the hydrothermal activity could have been buffered at near-neutral to alkaline conditions'.
- Lines 331-336: We followed on the reviewer comments to mention that the thermodynamic properties for REE hydroxyl species likely need to be revised.

Reviewer 2:

- All vocabulary suggestions, typos and mineral names have been corrected according to the reviewer's comments.
- Line 131-135: The presence of high amounts of calcite as an obvious indicator for not so acidic conditions is now mentioned.
- Lines 246-250: I completely agree with the reviewer's comment. The observed prograde solubility and 'Goldilocks effect' for La are here just a hint of what could occur for the LREE and more experiments are needed. Those were indeed beyond the scope of our study, where we were mostly interested in describing REE complexes in alkaline fluids and find a way to transport REE and F together at temperature above 100°C. I know that our study will encourage further experimental work on REE solubility using other technics, including from the reviewer's team! On our side, we foresee to conduct new experiments in diamond-anvil cell to overcome the detection limits we encountered in this study. I expect those experiments to take place sometimes in 2023-2024, due to 1) personal time limitations (!) and 2) time needed for proposal acceptance at the ESRF synchrotron.
- Lines 255-257: The sentence was reorganized to remove those double brackets.
- I chose not to implement the extra references suggested by the reviewer as there are already many field studies referenced and the reference list here provided is actually quite long compared to the journal's suggestion (90).

- Figures 1 and 2 axis have been changed back from commas to points.
- Figure 2 caption: The reviewer asked whether we could estimate La and Gd aqueous concentrations from the eH values. We can give a rough estimate based on comparison to standard solutions of REE in HCl: we estimate they are in the 100-1000ppm range and mention this briefly for La ('close to detection limit of 10 ppm' and '10 times higher' on lines 235-236) and Gd (400-600ppm at 200C on lines 240-241; only few tens of ppm at higher T, as mentioned on line 259). However, this approach remains very unprecise, as the fluorescence signal recorded here is impacted by the composition and density of the fluids and thus differs between our alkaline compositions and the HCl standard solutions. We thus preferred not to further insist on concentration but rather mention the concentration trends (ie., increases or decreases).
- Lines 314-317: The reasoning for our 'preferred' structure is mentioned in the previous line: we only fit 2 carbonate groups around Yb/Gd, hence the complex $[\text{REE}_3(\text{CO}_3)_2(\text{OH})_4]^+$ seems more likely than $\text{REE}_3(\text{CO}_3)_3(\text{H}_2\text{O})_{12}]^{3+}$. Furthermore, due to basic pH, we expect the presence of OH groups.
- Line 350: 'these' refers to our alkaline fluids.
- Lines 353-354: the idea would be that F enters the aqueous complexes in the place of OH. And it quickly precipitates as fluorcarbonates. As mentioned earlier in the text, this is however extremely difficult to prove as O and F have very similar bond distances.
- Line 369-372: Burbankite and carbocearnite are not mentioned here as they seem to form from more concentrated fluids according to your own study.
- Line 417-422: The sentence was reworked to better explain what the processes behind LREE enrichment and HREE removal.

- Figure 3 caption: the differences underlined by the dashed lines are now precised. They are differences in the shape and the position of the EXAFS oscillations.

Reviewer 3:

- As reviewers 1 and 2 did underline many typos, I hope most are now corrected!